



# Airborne measurements of oxygen concentration from the surface to the lower stratosphere and pole to pole

Britton B. Stephens[1], Eric J. Morgan[2], Jonathan D. Bent[2,1,4], Ralph F. Keeling[2], Andrew S. Watt[1], Stephen R. Shertz[1], and Bruce C. Daube[3]

[1]National Center for Atmospheric Research, Boulder, Colorado, USA
[2]Scripps Institution of Oceanography, La Jolla, California, USA
[3]Harvard University, Cambridge, Massachusetts, USA
[4]now with Picarro, Inc., Santa Clara, California, USA

*Correspondence to:* Britton B. Stephens (stephens@ucar.edu)

**Abstract.** We have developed in situ and flask sampling systems for airborne measurements of variations in the $O_2/N_2$ ratio at the part per million level. We have deployed these instruments on a series of aircraft campaigns to measure the distribution of atmospheric $O_2$ from 0–14 km and 87°N to 85°S throughout the seasonal cycle. The NCAR airborne oxygen instrument (AO2) uses a vacuum ultraviolet (VUV) absorption detector for $O_2$ and also includes an infrared $CO_2$ sensor. The VUV

5  detector has a precision in 5 seconds of $\pm$ 1.25 per meg (1 $\sigma$) $\delta(O_2/N_2)$, but thermal fractionation and motion effects increase this to $\pm$ 2.5–4.0 per meg when sampling ambient air in flight. The NCAR / Scripps airborne flask sampler (Medusa) collects 32 cryogenically dried air samples per flight under actively controlled flow and pressure conditions. For in situ or flask $O_2$ measurements, fractionation and surface effects can be important at the required high levels of relative precision. We describe our sampling and measurement techniques, and efforts to reduce potential biases. We also present a selection of observational

10  results highlighting the individual and combined instrument performance. These include vertical profiles, $O_2:CO_2$ correlations, and latitudinal cross sections reflecting the distinct influences of terrestrial photosynthesis, air-sea gas exchange, burning of various fuels, and stratospheric dynamics. When present, we have corrected the flask $\delta(O_2/N_2)$ measurements for fractionation during sampling or analysis, with the use of the concurrent $\delta(Ar/N_2)$ measurements. We have also corrected the in situ $\delta(O_2/N_2)$ measurements for inlet fractionation and humidity effects by comparison to the corrected flask values. A comparison of Ar/N$_2$-

15  corrected Medusa flask $\delta(O_2/N_2)$ measurements to regional Scripps $O_2$ Network station observations shows no systematic biases over 10 recent campaigns (+0.2 $\pm$ 8.2 per meg, mean and standard deviation, n = 86). For AO2, after resolving sample drying and inlet fractionation biases previously on the order of 10–100 per meg, independent AO2 $\delta(O_2/N_2)$ measurements over 6 more recent campaigns differ from coincident Medusa flask measurements by -0.3 $\pm$ 7.2 per meg (mean and standard deviation, n = 1361), with campaign-specific means ranging from -5 to +5 per meg.





# 1 Introduction

Atmospheric $O_2$ observations can be a powerful tool for elucidating carbon cycle processes on multiple time and space scales because of unique relationships between $O_2$ and $CO_2$ surface exchange (e.g., Keeling and Shertz, 1992; Stephens et al., 1998; Keeling and Manning, 2014; Ishidoya et al., 2013a; Nevison et al., 2015; Morgan et al., 2019b). Although measuring atmo-

spheric $O_2$ is challenging, because of the need to detect small variations against the large natural background, various in situ and flask-based techniques are now capable of achieving precision at the required $10^{-6}$ relative level (Keeling, 1988; Bender et al., 1994; Manning et al., 1999; Stephens et al., 2003, 2007a; Tohjima, 2000). Airborne measurements particularly have the potential to capture information on processes at large spatial scales, and to overcome uncertainty associated with the vertical mixing of flux signals away from the surface (e.g., Gerbig et al., 2003; Stephens et al., 2007b; Graven et al., 2013; Sweeney

et al., 2015).

However, aircraft pose significant limitations to instrument size, weight, and power, and are challenging platforms from which to conduct precise measurements. Cabin temperature can vary by 10°C and have local vertical gradients of 5°C m$^{-1}$, and cabin pressure can vary by 200 hPa. Furthermore, while profiling from the surface to 14 km in the tropics ambient humidity drops from over 30,000 to less than 20 ppm, ambient temperature drops by 85°C, and ambient pressure drops from 1000 to

150 hPa. To avoid fractionation of $O_2$ relative to $N_2$ or surface effects in the face of these and other challenges, it is generally necessary to actively control instrument flows, temperatures, and pressures, to dry the sample stream to a few ppm of $H_2O$, and to minimize the surface area and roughness of tubing experiencing temperature, pressure, and humidity changes (Keeling et al., 1998; Langenfelds, 2002). Additional sources of measurement bias may result from fractionation at sample inlets (Blaine et al., 2006; Steinbach, 2010; Bent, 2014), thermal diffusion of gases inside calibration cylinders (Langenfelds, 2002), and leaks of

cabin air reaching the inlet stream. Flask sampling reduces some of these challenges, because flasks can generally be sampled at higher flow rates, and the critical calibration and analysis steps all occur in a controlled laboratory environment. However, flask sampling may be subject to fractionation at the flask outlet during sampling (Bent, 2014) and storage effects (Steinbach, 2010; Keeling et al., 1998). In comparison, the advantages of in situ atmospheric $O_2$ measurements are the greatly increased spatial and temporal coverage and resolution, and the lack of sample storage concerns. Measurements of atmospheric $O_2$ have

been made on flasks collected from aircraft in a number of studies (Langenfelds, 2002; Sturm et al., 2005; Steinbach, 2010; Ishidoya et al., 2012, 2014; van der Laan et al., 2014; Bent, 2014).

Here we present an airborne in situ $O_2$ instrument that has flown on 14 campaigns since 2007 and an airborne flask sampling system that has flown on 17 campaigns since 1999 (Fig. S1 and S2). We focus on data from recent campaigns on the NSF/NCAR High-performance Instrumented Airborne Platform for Environmental Research (HIAPER) Gulfstream V (GV)

aircraft (UCAR/NCAR - Earth Observing Laboratory, 2005): the Stratosphere-Troposphere Analyses of Regional Transport campaign (START-08; Pan et al., 2010), five HIAPER Pole-to-Pole Observations campaigns (HIPPO 1–5, 2009-2011; Wofsy et al., 2011), and the 2016 $O_2/N_2$ Ratio and $CO_2$ Airborne Southern Ocean (ORCAS) study (Stephens et al., 2018); as well as the Airborne Research Instrumentation Testing Opportunity (ARISTO-2015) campaign on the NSF/NCAR C-130 (UCAR/NCAR - Earth Observing Laboratory, 1994) and four Atmospheric Tomography Mission (ATom 1–4, 2016-2018) campaigns on the



NASA DC-8. Selected results and methods from these instruments have been previously presented in Bent (2014), Resplandy et al. (2016), Nevison et al. (2016), Stephens et al. (2018), Asher et al. (2019), Morgan et al. (2019b), and Birner et al. (2020).

The in situ NCAR airborne oxygen instrument (AO2) measures $O_2$ concentration using a vacuum ultraviolet (VUV) absorption technique. AO2 is based on earlier shipboard (Stephens, 1999; Stephens et al., 2003) and laboratory instruments using

the same technique, but has been designed specifically for airborne use to minimize motion and thermal sensitivity, and with a pressure- and flow-controlled inlet system. The VUV detector in AO2 uses a low-pressure small-volume detector cell, which is possible due to the very high absorption cross section for $O_2$ in the VUV. The small cell allows rapid switching between sample and reference which, combined with the strong absorption, provide unparalleled signal to noise ratio and rapid time response. We tested an early prototype in situ instrument on the NSF/NCAR C-130 during the Instrument Development and

Education in Airborne Science (IDEAS-1 and IDEAS-2, 2002) campaigns. AO2 first made research quality measurements on the University of Wyoming King Air during the 2007 Airborne Carbon in the Mountains Experiment (ACME-07; Desai et al., 2011),

The NCAR / Scripps Medusa airborne flask sampler was designed to collect cryogenically-dried air samples under controlled pressure and flow conditions. The drying and pressure and flow control are necessary to minimize fractionation of the collected

air during sampling, and reduce surface effects from both the flasks and sample tubing. The Medusa flasks are maintained at 1 atm pressure and a few ppm of $H_2O$ at all times, from preparation and shipping through sampling and analysis. In addition, the flasks are contained in an insulated enclosure to minimize thermal fractionation effects during sampling. An earlier 16 flask version of the sampler flew on the University of North Dakota Citation II aircraft during the $CO_2$ Budget and Rectification and Airbone Study (COBRA-1999test, COBRA-2000 and COBRA-2003; Stephens et al., 2000; Kort et al., 2008) and during

IDEAS-1. This version also flew on the NSF/NCAR C-130 during ACME-04, but collected smaller samples for $^{13}C$ of $CO_2$ and not $O_2$ measurements (Sun et al., 2010). We repackaged Medusa for START-08 and then increased the sampling capacity to 32 flasks for HIPPO-1.

Here we describe the AO2 (Sect. 2) and Medusa (Sect. 3) configurations and operational procedures as flown during the most recent ORCAS and ATom campaigns, and list significant past configuration changes in Table S2. For AO2, we focus on aspects

specific to airborne deployment and other modifications from the instrument described by Stephens et al. (2003). Additional Medusa details can be found in Bent (2014). In Sect. 4, we discuss potential sources of measurement bias and our efforts to minimize them. We confine this discussion primarily to the $O_2$ measurements, and leave discussion of potential $CO_2$ biases for presentation elsewhere; for HIPPO $CO_2$ instrument intercomparisons see Santoni et al. (2014) and Gaubert et al. (2019). We then present a selection of measured vertical profiles, $O_2$:$CO_2$ correlations, and latitude / altitude cross-sections (Sect. 5) that

highlight the resolution of the measurements and their ability to distinguish the influences of specific processes.



## 2   NCAR Airborne Oxygen Instrument

### 2.1   Instrument description

The AO2 gas handling system depicted in Fig. 1 consists of a pump box, a cylinder box, an analyzer box, an inlet, and a cryotrap. See Table S1 for selected vendor and part numbers. We describe AO2 here in the general order of the sample air

moving through the system. During ATom 2–4, AO2 sampled from an aft-facing 3.2 mm OD, 2.2 mm ID electropolished stainless steel inlet inside of a HIAPER modular inlet (HIMIL) pylon, 30 cm from the aircraft skin. The HIMIL is a cigar-shaped tube with a 6.4 mm ID conical knife edge forward inlet, a 22 mm ID cylindrical bore, and a 9.5 mm ID outlet, and is designed to slow the relative air speed to minimize acceleration effects at the internal 3.2 mm inlet to the instrument. The HIMIL contains the AO2 and Medusa sample inlet tubing, an open tube connected to a pressure sensor, and an unused sample

tube. The AO2 inlet is the aftmost of these. Except where noted, all tubing exposed to sample air in AO2 is 3.2 mm OD, 2.2 mm ID electropolished Sulfinert-treated stainless steel, to minimize surface adsorption and desorption effects.

Immediately inside the aircraft from the inlet, a manual three-way valve selects air from either the inlet or a line purge cylinder. Directly following this selection, a proportional solenoid valve actively controls the pressure in the line to the instrument rack and at the inlet to an upstream vacuum pump. The pump uses teflon-coated diaphragms and teflon valve plates, sealed

by o-rings to custom aluminum heads, the latter used to minimize volume. Before entering the pump box, we filter sample air using a 3 μm pore by 47 mm diameter mixed cellulose ester filter in a stainless steel holder. The feedback controller for the inlet solenoid valve is referenced to an absolute pressure sensor downstream of the pump, and maintains a pump outlet pressure of $1050 \pm 7$ hPa ($1\,\sigma$ at 0.4 Hz) over the full range of flight altitudes. The sample air is then cryogenically dried with a 1.6 cm ID by 20.5 cm long electropolished stainless steel trap immersed in a dry ice and Fluorinert slurry at -78.5°C, to a depth of 18

cm at the start of a flight. At the pump outlet pressure of 1050 hPa this results in a saturation vapor concentration of 1.5 ppm. The air enters the trap at the top and exits through a 3.2 mm OD, 2.2 mm ID dip tube extending near the bottom of the trap. We use 3 mm glass beads in the lower 10 cm of the trap to minimize volume below the area of most ice accumulation and to restrict the free passage of any ice particles that might break loose, resulting in an approximate trap volume of 28 ml.

After compression and drying, the sample air is selected to either be measured or purged by a solenoid manifold that can

also select one of several calibration gases to be measured or purged. These calibration gases include a high span (high $O_2$ and low $CO_2$ concentration), a low span (low $O_2$ and high $CO_2$ concentration), a long term reference, and a working tank. All calibration gases are composed of ambient air, dried to less than 1 ppm $H_2O$. The working tank runs continuously as a reference, and can also be selected for measurement to be used as an additional $CO_2$ calibration; for $O_2$ the working tank is only used for diagnostic purposes as there is potential for fractionation in splitting the flow in the cylinder box manifold. The

calibration gases are contained in high pressure, 4.7 l fiber-wrapped aluminum cylinders, horizontally mounted in a block of foam insulation 5 cm thick at the outside walls. Two-stage brass regulators on each calibration gas cylinder are adjusted to match the delivery pressure of the inlet sample pump during preflight, but as they are referenced to cabin pressure, the absolute delivery pressure of the regulators varies in flight. The inlet line purge gas is in a similar cylinder, but mounted vertically outside of the insulated cylinder box, and uses a similar regulator but with a lower delivery pressure of 70 hPa above cabin. During



ACME-07, prior to Research Flight 10 on START-08, and during HIPPO-3, we used a coiled 6.4 mm diameter electropolished stainless steel moisture trap held at approximately 1°C as a preliminary drying stage. Out of concerns for surface effects, and because the aircraft spent the most time sampling very dry air, this trap is no longer used (Table S2).

The working tank and the selected sample or calibration gas, which we refer to as the span gas, are next pulled through the analyzer box by a downstream vacuum pump with sufficient capacity to operate the detector cell at < 100 hPa. Both air streams first pass (in sequence) an absolute pressure sensor, a proportional solenoid valve, a mass flow meter, and a 13 hPa full scale differential pressure sensor referenced to a common 500 ml insulated volume, maintained at 400 hPa. The feedback controllers for these solenoid valves actively match this pressure to within $\pm$ 1.5 Pa (1 $\sigma$ at 0.4 Hz). The span gas is then measured for $CO_2$ by a single cell non-dispersive infrared $CO_2$ / $H_2O$ sensor. We replaced the $CO_2$ sensor internal plastic tubing with 3.2

mm OD stainless steel, but to avoid ground loops through the sensor, we inserted Teflon union fittings inline to break electrical conductivity. The two streams are then cryogenically dried, the second time for sample gas and as a precaution for calibration and working tank gases, in 3.2 mm OD, 2.2 mm ID by 120 cm long coiled tubes immersed in the dry ice slurry. At 400 hPa, the saturation vapor concentration in these traps is 3.8 ppm $H_2O$. While this is wetter than saturation for the first stage trap and calibration gas, we include it because the first stage sample trap may not dry to saturation owing to residence time or diffusion

limitations, and the span gases may pick up a small amount of water permeating through the seals in the $CO_2$ sensor (see Sect. 4.5 below). The second stage span gas trap is intended to remove any of this water, and we include a trap on the working tank line both for consistency and as an added precaution.

   After the coiled tube traps, the gases reach a changeover valve manifold, with two rapid-switching, long-life, miniature 3-way solenoid valves, configured to work as a low-volume 4-way changeover valve. These valves alternately select the span

or working tank gas to either be measured by the VUV detector cell or vented through a bypass line. On the VUV detector line, a 0.2 mm sapphire jewel orifice immediately upstream of the cell acts as a critical flow orifice and reduces the pressure to 95 hPa as it passes through the cell. A proportional solenoid valve downstream of the cell controls this pressure to within $\pm$ 0.009 Pa (1 $\sigma$ at 0.4 Hz), by referencing a 1.3 hPa full scale differential pressure sensor to a second 500 ml insulated volume. Between the changeover valve and the orifice we use 1.0 mm ID tubing to minimize sweepout times. A manual needle valve is

located on the bypass line to match the combined flow impedance of the VUV detector line. The flow through the instrument is nominally 100 sccm, set by the sapphire orifice and the upstream reference pressure. Solenoid valves and pressure gauges allow the reference volume pressures to be monitored and adjusted if necessary between flights or for testing.

   The VUV source consists of a Xe resonance lamp with a $MgF_2$ window powered by a 15 W 180 MHz radio frequency oscillator, which emits strongly at 147 nm and more weakly at 129 nm (Okabe, 1964). The detector is a CsI photocathode

with a $MgF_2$ window and peak output of 100 nA. The analyzer box and $O_2$ sensor have essentially the same configuration as the shipboard instrument described in Stephens (1999) and Stephens et al. (2003), with a few key differences. The airborne instrument, and our laboratory system, now use a sealed Xe lamp instead of the original flow through design. In addition to the $MgF_2$ windows on the lamp and detector, we have also employed a sapphire window in front of the detector on the aircraft instrument to eliminate the secondary Xe line at 129 nm. Earlier tests, using a sapphire window fused to the lamp body

with a proprietary coating, showed large humidity effects that Stephens et al. (2003) speculated might have resulted from water



adsorption on the sapphire. These effects no longer appear to be as significant, either because of the use of a non-fused sapphire window or because the earlier problems may have been a result of inadequate drying. However, to further minimize concerns we place an additional $MgF_2$ disc on the sample cell side of this sapphire disc. We also use a 1 mm thick aluminum aperture disc with a 6.4 mm diameter hole between this $MgF_2$ window and the sample cell to avoid damaging the CsI photocathode

with too much light. The VUV absorption cell is thus defined on one side by the detector and aperture disc, and on the other by the lamp window, and is a 4.3 mm long by 13 mm diameter cylinder with a 5.3 mm path length accounting for the aperture.

Fig. 2 shows the relationship between detector voltage and cell pressure with this configuration, along with predicted noise contributions from thermal and shot noise. Using sapphire to exclude the 129 nm line in a region of weaker $O_2$ absorption allows the instrument to be run at higher cell pressures and greater absorption factors. The 95 hPa cell pressure corresponds to

an optical depth of 3.8, or absorption of 98 % of the light, and a scaling factor of 3.0 between changes in $\delta(O_2/N_2)$ and relative changes in signal ($\Delta V/V$). This factor differs from 3.8 because the conversion between relative changes in mole fraction and $\delta(O_2/N_2)$ includes division by $(1-X_{O_2})$ (see eq. 4 in Keeling et al., 1998). We amplify and convert the resulting detector current of approximately 80 nA with a low noise op-amp and $1.25 \times 10^8$ ohm resistor (Stephens et al., 2003), and measure it using a 24-bit analogue to digital converter. As indicated in Fig. 2, there is a trade off between absorbing more light to achieve

greater sensitivity and the increase in shot noise with the reduced number of photons reaching the detector, and also current limits defined by the photocathode and op-amp configuration. Fig. 2 also shows the typical noise from the instrument running calibration gas without switching in lab, which indicates that the detector is within a factor of 2.4 of the shot and Johnson noise limit, and that the predicted noise is not particularly sensitive to the choice of cell pressure between 80 and 120 hPa.

Our aircraft and lab systems also do not have the beam-splitter and second detector described in Stephens et al. (2003)

because we found that measurement noise could not be reduced by referencing to the unabsorbed beam, either because lamp output is not a dominant source of noise or plasma variations were imaged differently by the two detectors. We correct for the imperfect control of cell pressure on short timescales using the measured pressure differential between the cell and reference volume. We also employ a second identical 1.3 hPa full scale differential pressure sensor with both ports plumbed together to correct for acceleration effects on the primary sensor (Fig. 1). These sensors have acceleration sensitivities of approximately

0.1 Pa s$^2$ m$^{-1}$. We orient all pressure sensors parallel to the longitudinal axis of the aircraft, to minimize the impact of vertical and horizontal accelerations during turbulence, but they do experience changes in the longitudinal component of gravity with aircraft pitch, and longitudinal accelerations during intentional yaw maneuvers, or on take-off or landing (Sect. 4.6.5).

AO2 control and data acquisition is by an embedded computer and analogue to digital converters in each box. In addition to the primary sensor measurements, for diagnostic purposes, AO2 logs 16 temperatures, 12 pressures, and 4 flows at 0.4 to 10

Hz.

## 2.2   Measurement approach and precision

To achieve the high levels of precision desired, AO2 switches between sample gas and working tank gas approximately every 2.3 s, more than a factor of 2 faster than the earlier shipboard instrument (Stephens et al., 2003). The AO2 measurement is then based on the amplitude of the resulting square wave as defined by the bidirectional difference in signals between a particular





jog and an average of the prior and subsequent jogs. This yields a statistically independent measurement every 4.6 s (hereafter rounded to 5 s), though we report partially overlapping differences every 2.3 s. The switching time is set by the amount of time the instrument needs to record 20 detector voltages at 10 Hz and then housekeeping variables between switches. Fig. 3 shows the 5 s square wave signal averaged over calibration and sample periods under different drying conditions. The low

volume of the switching solenoid valves, the detector cell, and the intervening tubing, and low cell pressure result in very fast cell flushing times on the order of 0.02 s. The instrument records a number of housekeeping signals over the first 0.3 s after switching and by the time it records the VUV detector signal again, the cell has almost completely swept out. As a result, artifacts due to incomplete sweepout are small. The difference in slopes between the working tank and span segments of the square wave is only marginally influenced by the magnitude of the difference in concentration between the two gases, and we

do not exclude any data following the changeover valve switch. However, the slope difference between working tank and span segments can provide an important diagnostic of several other possible issues, including delays in pressure equilibration from small cross-port leaks in the changeover valve or differences in humidity or hydrocarbons leading to chemical interactions with optical surfaces under intense VUV (see Sect. 4.5 below).

Atmospheric oxygen is quantified as the ratio of the relative abundance of $O_2$ to the relative abundance of $N_2$ in units of per

meg (Keeling and Shertz, 1992; Keeling and Manning, 2014), *i.e.*,

$$\delta(O_2/N_2) = \left( \frac{(O_2/N_2)_{sample}}{(O_2/N_2)_{ref}} - 1 \right) \times 10^6 \tag{1}$$

where one per meg represents a one-millionth change in the $O_2/N_2$ ratio relative to an arbitrary reference. For the Scripps $O_2$ Program $O_2$ scale this reference is a suite of high-pressure cylinders maintained at Scripps. An analogous definition to Eq. 1 is used to report measurements of the $Ar/N_2$ ratio. In addition to $\delta(O_2/N_2)$ and $CO_2$, we also report values for the derived tracer

atmospheric potential oxygen (APO; Stephens et al., 1998):

$$APO = \delta(O_2/N_2) + \frac{1.1}{X_{O_2}} (CO_2 - 350) \tag{2}$$

where 1.1 is the estimated stoichiometric ratio of long-term terrestrial biosphere $O_2$ and $CO_2$ exchange, and $X_{O_2}$ is the mole fraction of $O_2$ in dry air as defined by the Scripps $O_2$ Program $O_2$ Scale. APO is designed to be conservative with respect to terrestrial photosynthesis and respiration, to only have a small fossil-fuel sink, and to primarily reflect $\delta(O_2/N_2)$ and $CO_2$

exchange with the oceans. Although 1.05 is likely a better $O_2{:}CO_2$ ratio for canceling short term terrestrial influences (Stephens et al., 2007a; Battle et al., 2019), we continue to use 1.1 here for consistency with past studies, and encourage sensitivity tests over a range of possible $O_2{:}CO_2$ ratios.

Fig. 4 shows instrument noise as a function of hypothetical switching time, based on 40 min of calibration gas analysis with no actual valve switching in lab. This figure includes the 2-sample Allan standard deviation, as well as the standard deviation

of the bidirectional 3-sample differences we use, the latter of which has a broad minimum of 1.25 per meg between simulated switching times of 2 and 3 s. The slope of the rise in noise for shorter intervals suggests that there would be no improvement from switching faster. Fig. 4 also shows typical noise values from the instrument while running calibration gas with switching, and while measuring ambient air with stable concentration, both during field conditions. Although at times AO2 noise while





running calibration gas is 1.25 per meg or better (1 $\sigma$ in 5 s) the value shown here of 1.6 per meg is more consistently achievable. For example, the median of the 5 s noise levels within all individual calibration gas intervals was 1.5 per meg for ATom-3, and 1.7 for ATom-4. Our correction of imperfect cell pressure control based on the downstream 1.3 hPa differential pressure sensor has a negligible effect for the lab conditions shown in Fig. 4, but can reduce the noise in turbulent flight conditions by a factor

of 5 or more.

We find similar noise levels whether switching or not switching the changeover valve between working tank and a calibration gas, or between working tank and span gas when the trap is warm, indicating that pressure and flow fluctuations from the actual switching do not add noise. Rather, the slight increase in noise for calibration gases in flight relative to lab conditions is likely a result of aircraft motion and slight drift within the flight calibration intervals. However, the switching of the changeover valve

does introduce extra noise when running either long-term surveillance gas or sample air through the first stage trap when it is cold, suggesting an interaction between flow perturbations and thermal diffusion in the trap (Keeling et al., 1998). Thus, our typically achieved precision when measuring ambient air in stable conditions is 2.5–4.0 per meg, 1 $\sigma$ in 5 s. Fig. 5 shows $O_2$ and $CO_2$ signals over the course of an entire flight, including several hours of preflight and 15 min of postflight inlet line purge analysis. For the 36-min high altitude period between 22:23 and 23:00 in Fig. 5, the variability in $\delta(O_2/N_2)$ is $\pm$ 3.5 per meg (1

$\sigma$ for 5 s samples), and as low as $\pm$ 2.5 per meg for similar periods on other flights (e.g. Fig. S3). For comparison, 2.5 per meg is equal to a change of 0.4 ppm in $O_2$ mole fraction or the addition of 0.5 micromoles of $O_2$ to 1 mole of air (Keeling et al., 1998; Kozlova et al., 2008). This variability averages as white noise, and with statistically independent samples every 5 s, the precision on a 1-min average is approximately $\pm$ 0.7–1.1 per meg while measuring ambient air and 0.4 per meg for calibration gas.

## 2.3    In-flight calibration strategy

The AO2 high pressure reference cylinders are equipped with brass valve manifolds sealed with silver-coated c-rings. These manifolds have needle valves going to either a fill and laboratory analysis port or a regulator for use in flight, a burst disc, and a 20 cm dip tube to minimize thermal fractionation in withdrawn air (Keeling et al., 2007). These dip tubes have been either 3.2 mm OD, 1.4 mm ID nickel or 0.16 mm OD, 1.0 mm ID electroformed nickel supported by a perforated 6.4 mm electropolished

stainless steel tube. We fill these cylinders to 200 bar with ambient air from larger cylinders filled by NOAA/GML at Niwot Ridge, CO. We then adjust the $O_2$ and $CO_2$ concentrations in these cylinders, and calibrate them against laboratory references traceable to both the Scripps $O_2$ Program $O_2$ scale, as defined on 16 March 2020, and the WMO X2007 $CO_2$ scale. We measure each flight cylinder in lab over a period of several weeks both before and after a field deployment to detect and account for any drift (Sect. 4.6.1).

Approximately every 50 min during flight, we measure the high and low span cylinders for 2.5 min each, alternating their order each time to detect any flushing issues (Fig. 5). We measure the working tank against itself for 2.5 min for an additional $CO_2$ calibration point every other calibration cycle, and we measure the long term reference for 2.5 min as a system diagnostic every third calibration cycle. Before each calibration gas is measured, we purge it at our sample flow of 100 ml min$^{-1}$ for 2.5 min. We exclude 60 s of data after every switch to a calibration gas or back to sample and average the remaining data for each





calibration period. Allowing for these transitions, a 2-point calibration excludes 6 minutes of ambient air measurement and a 4-point calibration 11 minutes.

Starting with HIPPO-3, we added a fifth cylinder of air to purge the inlet line during pre and post flight periods, to prevent ingesting aircraft exhaust and to dry inlet lines during preflight (Sect. 4.3 and Sect. 4.5). Starting with ORCAS, we also used

this purge air to flush and dry inlet tubing during maintenance days. This cylinder is mounted vertically and external to the insulated cylinder box. We load dry ice in the dewar approximately 2.5 hours before takeoff, and then start the working tank and inlet line purge gases flowing and turn on the VUV lamp 2 hours before takeoff, to warm up and dry out the instrument. During the hour immediately before takeoff, we run our calibration sequence 4 times at 15 min intervals to flush the regulators and dry out the calibration manifold and tubing.

## 2.4  Data processing

Similar to the processing of shipboard data described in Stephens et al. (2003), the amplitude of the 5 s switching signal is proportional to the mole fraction of $O_2$ in the sample gas and forms the basis of the measurement (Fig. 5).

For $O_2$ we calculate a linear fit for each paired high span / low span calibration cycle to apparent mole fraction (Stephens et al., 2003) and interpolate these fit parameters linearly in time to ambient air or long term reference measurements. For $CO_2$

we first calculate a quadratic fit to calibration cycles that also include the working tank gas and interpolate and apply these parameters to cycles with only high span and low span gases, before calculating secondary linear fits for these cycles and interpolating to sample and long term reference measurements. After calculating apparent mole fractions of $O_2$ for the entire flight, we convert these to $\delta(O_2/N_2)$ and correct for dilution using the concurrent calibrated measurements of $CO_2$ (see eq. 4 in Stephens et al., 2003). After the preflight period the $O_2$ calibration as defined by the linear fit to high and low span cylinder

measurements is very stable, with typical drift rates of less than 5 per meg per hour and less than 15 per meg over an 8–10 hour flight. For Fig. 5, the standard deviation of the 13 in-flight high span calibration gas measurements is $\pm$ 2.5 per meg. We shift our measurements in time using inlet lags empirically determined from the switches to and from inlet line purge cylinder gas before and after each flight, plus a minor additional pressure dependent lag for the remaining portion of the inlet upstream of this valve. At our present sample flow rate and trap volume, the total inlet lag is approximately 50 s with a $\pm$ 10 s smoothing

window attributable to shear and turbulence induced mixing in the tubing and traps. This lag time was 10 s shorter with the small diameter trap used prior to ARISTO-2015, and has varied by a similar amount owing to differences in flow and inlet lengths across campaigns.

On specific flights and campaigns, we make additional corrections to the AO2 data as described in Sects. 4.2, 4.3, 4.4, and 4.5.





## 3 NCAR / Scripps Medusa Flask Sampler

### 3.1 Sampler description

The Medusa gas handling system is shown in Fig. 6. Medusa consists of two identical insulated flask boxes, each of which holds 16 flasks, a control box which houses most of the pressure control system and the system computer, a valve box which
holds two additional multi-position gas handling valves, two external pumps, an inlet, a purge cylinder, and a stainless steel dewar. See Table S1 for selected vendor and part numbers. Medusa shares the same HIMIL pylon with AO2 but samples from a 6.4 mm OD, 4.6 mm ID aft-facing electropolished stainless steel inlet tube, which is in the second position behind a fourth unused inlet tube of the same size. Air is drawn from the inlet into the system by an upstream vacuum pump modified as for AO2, while a pressure controller located immediately inside the aircraft maintains a constant pressure upstream of the pump.
Similar to AO2, between the inlet and this first control valve, a manual 3-way valve allows the system to sample from a purge gas cylinder containing dried natural air. In the case of Medusa, this inlet purge cylinder was added prior to ORCAS. Between the pressure controller and the pump, the sample air is filtered by a 30 μm pore by 47 mm diameter polypropylene filter in a stainless steel holder. After passing through the upstream pump, the sample air is dried in series by two 2.2 cm ID by 23 cm long electropolished stainless steel traps immersed in a similar dry ice and Fluorinert slurry as for AO2. We actively heat the
inlet to the upstream trap to a temperature of 4.5 °C to prevent water freezing out before reaching the trap itself and obstructing flow. When the ambient dewpoint is greater than 4.5 C, water will start to condense at this point, but as it is only several cm directly above the trap we expect any formed drops to migrate into the cold trap. The dried air is then directed to the inlet of one of the 32 flasks by a series of rotary multiposition valves. A second pump and pressure controller downstream of the flask outlets controls flask pressure to approximately one atmosphere. Medusa includes a single-cell $CO_2$ / $H_2O$ sensor downstream
of the flasks to provide diagnostics of flask drying and mixing, and detection of potential cabin-air leaks. All Medusa tubing is electropolished 6.4 mm OD, 4.6 mm ID stainless steel except for approximately 50 cm flexible lines of 6.4 mm OD, 4.3 mm ID ethylene copolymer lined tubing upstream and downstream of each flask. Medusa collects air samples into 1.5 l (30 cm long by 8 cm diameter) borosilicate glass flasks that are contained within a block of foam insulation 3 cm thick at the outside walls, with the stopcocks and tubing connections protruding from the foam. The sample air enters the flask at the exposed end
and exits the flasks via a 24 cm dip tube extending into the flask, with the intent to minimize thermal fractionation effects and improve flushing (see Sect. 4.1). The stopcocks use Viton o-rings lightly coated with vacuum grease to minimize permeation effects and flask breakage (Keeling et al., 1998).

 The flow rate through the system is set by the upstream and downstream pressure set points, selected from a range of predetermined options to maximize flow while maintaining approximately 1 atm in the flask and the upstream controller
pressure set point below ambient. At the maximum altitudes of the GV and DC-8, we typically use an upstream set point of 146 hPa with a flow rate of approximately 1550 ml min$^{-1}$, and at the lowest altitudes we typically use an upstream set point of 226 hPa which produces a flow rate of approximately 2700 ml min$^{-1}$. After switching between pressure settings, we allow 10 minutes for the system to stabilize before sampling when switching to the lowest flow, and 5 minutes when switching to the highest flow.





Before each campaign, we purge the flasks in the laboratory with 5 volumes of cryogenically dried cylinder air at ambient $CO_2$ and $O_2$ levels. and store them with an internal pressure of 1 atm. Then before each flight we purge them for one minute on the high flow setting in the sampler with cylinder air. Before sampling we flush the flasks with a minimum of 5 volumes of sample air before moving to the next position. At the time of sample collection, the system temporarily switches flow to a

bypass loop with matched impedance, while a rotary valve isolates the air within the sampled flask and connecting tubing. At a later time, nominally several minutes to an hour, the onboard operator manually closes the flask stopcocks to preserve the sample. We record system pressures, flows, and diagnostic $CO_2$ and $H_2O$ at 1 Hz. After each flight we disconnect and remove the upstream trap, using normally closed quick connect fittings. During maintenance days, we exchange flasks, install a dry upstream cryotrap, and pressure test all connections for leaks.

## 3.2 Flask analysis

Sampled flasks are shipped to Scripps Institution of Oceanography for analysis. We try to minimize the temperature range to which flasks are exposed before analysis, but this is not always possible, and hence flasks can be exposed to a variety of environments before they reach the lab. Flasks are typically analyzed within 3 months of collection, with a median storage time of all flasks of 80 days. The flask analysis includes measurements of $CO_2$ on a non-dispersive infrared analyzer (LI-

COR 6252), and of $\delta(O_2/N_2)$ and $\delta(Ar/N_2)$ on a sector-magnet mass spectrometer (Micromass IsoPrime; Keeling et al., 2004), followed by extracting $CO_2$ for subsequent isotopologue measurements. We do not find any evidence of storage effects in $\delta(O_2/N_2)$, $\delta(Ar/N_2)$, or $CO_2$ over these time scales. We measure the extracted $CO_2$ for $^{13}C$ and $^{18}O$ of $CO_2$ on an Optima mass spectrometer (Guenther et al., 2001), and recapture and preserve the $CO_2$ for eventual measurement of $^{14}C$. The single-flask 1 $\sigma$ precision for the $\delta(O_2/N_2)$ and $CO_2$ measurements are $\pm$ 3 per meg and $\pm$ 0.15 ppm, respectively (Keeling et al., 2004).

The $CO_2$, $\delta(O_2/N_2)$, and $\delta(Ar/N_2)$ measurements are done by first withdrawing 150 ml of air over 5 minutes out of the flask while replacing this lost volume with a purge gas of known concentrations and artificially low $CO_2$. The flasks are contained in an insulated box during analysis to minimize temperature fluctuations. If the flasks are to be measured subsequently for stable carbon isotopes, we correct for the dilution with purge gas by remeasuring the $CO_2$ content after the flasks have equilibrated overnight (Kort et al., 2008; Bent, 2014). This second analysis is done on a 90 ml subsample without replacement.

Calibration gases for the mass spectrometer are introduced from the laboratory interferometer system via a tee, and we correct the Medusa flask measurements for empirically determined offsets owing to fractionation at this tee of +6.4 and +8.5 per meg for $\delta(O_2/N_2)$ and $\delta(Ar/N_2)$, respectively. The direction of flow during analysis is in through the dip tube to be consistent with Scripps $O_2$ Program network flasks. The flasks are mounted horizontally during analysis with half of the flasks having their dip tubes upwards and the other half rotated 180 degrees with their dip tubes downwards. We are able to detect gravitational

fractionation effects during analysis with flasks with lower outlets having enhanced concentrations of the heavier species, and vice versa. We apply empirical corrections for this effect of $\pm$ 0.8 and $\pm$ 2.4 per meg for $\delta(O_2/N_2)$ and $\delta(Ar/N_2)$, respectively.



## 3.3 Data processing

Laboratory tests indicate that mixing of air in the flasks during sampling is well approximated by an e-folding time equal to the flask volume divided by the flow rate. The characteristic mixing time for the Medusa flasks is approximately 33 seconds at the highest flow rate and 60 seconds at the lowest flow rate. We estimate the combined inlet and cold trap lag time for Medusa

from volume / flow to be 7 s at the highest flow setting and 13 s at the lowest flow setting. To compare the flask measurements to state parameters and other chemical measurements sampled at higher frequency, we use a weighting kernel for each flask that is based upon the measured flow rate, tubing lags, and sampling start and end times (Kort et al., 2008; Bent, 2014):

$$w(t) = e^{-\left(\frac{t_f - t}{\tau}\right)} \tag{3}$$

where $w(t)$ is the weighting of any 1-second time increment t between the switch to the sampled flask and the switch to the next

flask, $t_f$, and $\tau$ is the flushing time in seconds, *i.e.*, the flask volume divided by the mean flow during the sampling period. $w(t)$ is scaled so that it sums to 1 for all non-missing values over a given sampling interval. These weighting kernels are reported along with the final Medusa data.

Despite careful attention in the field and lab, it is possible for flasks to experience leaks during the various stages of sampling, shipping, storage, and analysis. The analysis system at Scripps has automated checks to reject flasks with obvious anomalies

in fill pressure or other parameters. In addition, we manually identify and flag flasks with $CO_2$, $\delta(O_2/N_2)$, and $\delta(Ar/N_2)$ measurement well out of range of expectations from the concurrent $CO_2$ measurements made by AO2 and other instruments, $\delta(O_2/N_2)$ from AO2, and background $\delta(Ar/N_2)$ values. Of the 4,004 flasks sampled in the 12 most recent campaigns, 209 were flagged during analysis at Scripps and an additional 109 were manually flagged after analysis.

All flask measurements are referenced to a hierarchy of calibration cylinders to measure them on the Scripps $O_2$ Program $O_2$

and $CO_2$ scales (Keeling et al., 1998, 2007). The NCAR primary cylinders, measured both by Scripps and NOAA, allow us to establish a link between the Scripps $O_2$ Program $CO_2$ scale and the WMO $CO_2$ scale, in order to report the flask measurements on the WMO scale in campaign merge products, and to use common scales in comparison to AO2 measurements.

## 4    Discussion of potential sources of bias

Making measurements at the $10^{-6}$ relative level is challenging, and the developments of AO2 and Medusa have included

discovering and resolving a series of potential measurement artifacts, as described in the sections below. While we now have established practices to eliminate or minimize all of these effects, in some cases it has been necessary to develop empirical corrections for recognized systematic biases. The most significant of these effects have been those associated with inlet fractionation (Sect. 4.2) and inadequate drying of AO2 sample air (Sect. 4.5). We have also identified more subtle effects associated with thermal fractionation in Medusa flasks (Sect. 4.1), regulator and tubing surface conditioning (Sect. 4.3), and systematic

differences between measurements on climbs and descents (Sect. 4.4), When necessary, we calculate adjustments for bias effects as described in Sect. 4.1.1, 4.2.1, 4.3.1, 4.4.1, and 4.5.1. We discuss other potential sources of bias that have fortunately not required any adjustments in Sect. 4.6, and independent checks on measurement bias in Sect. 4.7. In all reported AO2 and



Medusa data products, we include both the raw and adjusted measurements, to support assessment of their impacts on various conclusions and use of the unadjusted data when that is more appropriate.

## 4.1 Fractionation of flask samples

Thermal or pressure-driven diffusive gradients can play a role in separating Ar, $O_2$, and $N_2$ under various conditions (Keeling
et al., 1998). This is a concern for flask sampling if temperature gradients exist at the point where molecules are committed to exiting the flask, e.g. at the dip tube tip when flowing out through the dip tube or at the stopcock if flowing in through the dip tube. Fig. S4 shows vertical profiles of $\delta(Ar/N_2)$ measurements on Medusa flasks from all campaigns. Observed decreases in $\delta(Ar/N_2)$ in the stratosphere are consistent with estimates of gravitational diffusion (Ishidoya et al., 2008, 2013b; Bent, 2014; Birner et al., 2020), but we expect vertical $\delta(Ar/N_2)$ gradients in the troposphere to be constant within a few per meg
(Bent, 2014). During HIPPO, the tropospheric $\delta(Ar/N_2)$ scatter about the campaign mean vertical gradient of approximately $\pm$ 20 per meg (1 $\sigma$) is considerably larger than the synoptic or spatial variations observed at surface stations (Keeling et al., 2004) and we attribute most of this to fractionation at the flask outlets. Fig. S5 shows Medusa flask $\delta(Ar/N_2)$ versus APO for each campaign along with reference lines for expected slopes for thermal and pressure fractionation of flask samples at 1 atm (Keeling et al., 2004). Fig. S6 similarly shows Medusa flask $\delta(Ar/N_2)$ versus normalized Medusa-AO2 APO differences. The
sign of the tropospheric Medusa $\delta(Ar/N_2)$ versus Medusa-AO2 APO difference relationship during START-08, HIPPO, and ORCAS on the GV suggest a fractionation effect on Medusa samples. Other evidence of much larger inlet fractionation for AO2 than Medusa (Sect. 4.2) suggests this is less likely an inlet effect and more likely a flask sampling effect. Though some of the larger $\delta(Ar/N_2)$ excursions are correlated with APO in Fig. S5, the Medusa-AO2 APO differences and $N_2O$ values in Fig. S6 suggest these are of natural stratospheric origin (Birner et al., 2020).

Although we cannot exclude a contribution from pressure-driven fractionation, because pressures are actively controlled and more consistent during sampling, we conclude that thermal gradients in the flasks are the most likely cause of this scatter. During HIPPO, $\delta(Ar/N_2)$ scatter was greater for flasks collected in the Medusa box closer to the GV cabin air vents. These flasks also had a lower mean $\delta(Ar/N_2)$ values (Bent, 2014). If this were a thermal fractionation effect, it could result from the flask dip tube being cold in comparison to the surrounding flask air. On the DC-8 during the ATom campaigns, when the
Medusa rack was inboard and away from any air conditioning vents, the $\delta(Ar/N_2)$ scatter was reduced by a factor of 2 (Fig. S4, S5, and S6) and the lower box difference was eliminated. On the GV during ORCAS, we added a metal plate to the side of the rack to partially block the cold cabin air vents, and the $\delta(Ar/N_2)$ scatter was reduced but the lower box still had a low bias. Also, during HIPPO-1 as a test we connected half of the flasks with flow into rather than out of the dip tubes, and these flasks showed greater scatter in $\delta(Ar/N_2)$. It is also possible that the use of the Medusa inlet line purge cylinder starting with ORCAS
contributed the reduction in $\delta(Ar/N_2)$ scatter during ORCAS and ATom, by reducing surface effects associated with drying of tubing early in flight.



### 4.1.1   Adjustments for thermal fractionation

We correct the Medusa $\delta(O_2/N_2)$ measurements for apparent thermal fractionation effects, by adjusting the measured $\delta(O_2/N_2)$ values according to the expected relationship with $\delta(Ar/N_2)$ for thermal fractionation, and report both original and corrected values. This correction is similar to that done in previous studies (Battle et al., 2006; Steinbach, 2010; Ishidoya et al., 2014; Bent, 2014). We use a constant reference value of 15 per meg $\delta(Ar/N_2)$ in the troposphere which is the approximate global surface annual mean from the Scripps $O_2$ Program network (Table S3). In the stratosphere, we adjust this reference $\delta(Ar/N_2)$ value by a linear fit between $\delta(Ar/N_2)$ and detrended $N_2O$, a proxy for age of air, with an intercept at the tropopause transition (Bent, 2014). We use $N_2O$ measurements from the Harvard Quantum Cascade Laser Spectrometer (QCLS, Santoni et al., 2014) and the NOAA PAN and Trace Hydrohalocarbon ExpeRiment (PANTHER, ATom-1 only) instruments. The corrected $\delta(O_2/N_2)$ values are then defined as:

$$\delta(O_2/N_2)^* = \delta(O_2/N_2) - \frac{\delta(Ar/N_2) - \delta(Ar/N_2)_{ref}}{3.77} \tag{4}$$

and by extension:

$$APO^* = \delta(O_2/N_2)^* + \frac{1.1}{X_{O_2}}(CO_2 - 350) \tag{5}$$

establishing new tracers, $\delta(O_2/N_2)^*$ and $APO^*$, that are largely insensitive to thermally-induced fractionation of flask samples.

This approach also corrects for any thermal fractionation effects during analysis, which would be indistinguishable from those during sampling. Adjusting to a constant tropospheric vertical profile in $\delta(Ar/N_2)$ also mostly compensates for potential inlet fractionation effects as discussed in Sect. 4.2. Fig. 7 shows the vertical distribution of the original $\delta(O_2/N_2)$ and corrected $\delta(O_2/N_2)^*$ values from Medusa flasks during all campaigns. The median $\delta(Ar/N_2)$ offset from the combined tropospheric value and stratospheric $N_2O$ relationship was -8.3 $\pm$ 26.4 per meg (1 $\sigma$), resulting in a median adjustment to Medusa $\delta(O_2/N_2)$ of +2.2 $\pm$ 7.0 per meg. For the three campaigns ATom 2–4, the median $\delta(O_2/N_2)$ adjustments are +1.9 $\pm$ 3.6 per meg (Table S3).

This method of correcting for $\delta(O_2/N_2)$ fractionation effects using $\delta(Ar/N_2)$ ignores the real boundary-layer seasonal cycles in $\delta(Ar/N_2)$ of up to $\pm$ 10 per meg amplitude and likely annual mean meridional $\delta(Ar/N_2)$ gradients on the order of 5 per meg (Keeling et al., 2004; Battle et al., 2003; Bent, 2014). Thus, scientific applications of Ar-corrected Medusa data, or of versions of AO2 data that are adjusted based upon comparison to Ar-corrected Medusa data, need to consider these additional influences. For example, Bent (2014) removed the estimated contribution of seasonal Ar variations to $APO^*$ from AO2 in an analysis of Southern Ocean $O_2$ exchange and Resplandy et al. (2016) used latitudinal gradients of APO from AO2 adjusted to non-Ar-corrected Medusa data in order to constrain interhemispheric ocean heat exchange.

### 4.2   Inlet fractionation

Pressure gradients also have the potential to diffusively separate $O_2$ and Ar relative to $N_2$ at aircraft inlets (Keeling et al., 1998). Steinbach (2010) proposed a model for aircraft inlet fractionation resulting from pressure gradients perpendicular to streamlines at the inlet, whereby forward facing inlets would preferentially sample heavier molecules at high aircraft velocity,



and vice versa for aft facing inlets. We expect diffusive fractionation effects to be greater by a factor of 3 for $\delta(Ar/N_2)$ than for $\delta(O_2/N_2)$ because of the greater mass difference (Keeling et al., 1998, 2004). Aircraft inlet fractionation will likely also be dependent on other factors such as ambient pressure, ram pressure, inlet flow rate, inlet tubing wall thickness, and angle of attack. We assessed these effects with several different configurations of aircraft and inlet design, starting with COBRA test

flights in 1999. These have included sampling at variable speeds, angles of attack, and flow rates, and switching between aft and forward inlets during stable flight conditions. These tests for COBRA using a forward facing 9.5 mm inlet tube, during the IDEAS campaign switching between forward and aft 3.2 mm and 6.4 mm diameter inlet tubes, and for START-08 or HIPPO using a HIMIL were inconclusive. However, vertical gradients in $\delta(Ar/N_2)$ (Fig. S4) and in AO2-Medusa $\delta(O_2/N_2)$ differences (Fig. 8) do suggest non-negligible inlet fractionation for the HIMIL. Furthermore, during ORCAS and ATom-1 when using aft

and side facing non-diffusing inlets, we observed much more dramatic fractionation effects.

One-dimensional model calculations of the balance between gravitational separation and tropospheric mixing suggest the mean vertical distribution of $\delta(Ar/N_2)$ should be constant to within 2 per meg below 9 km (Bent, 2014). Furthermore, marine boundary layer seasonal variations in $\delta(Ar/N_2)$ are relatively small at 10–20 per meg amplitude (Keeling et al., 2004) and would lead to vertical gradients of opposite sign in different seasons. We suspect the consistently observed tropospheric $\delta(Ar/N_2)$

gradients of approximately -2 per meg km$^{-1}$ for the HIPPO campaigns and +2 per meg km$^{-1}$ for ATom-1 shown in Fig. S4 resulted from inlet fractionation. The sign of the HIPPO gradients combined with the greater relative air speeds at higher altitude suggest preferential sampling of lighter $N_2$ molecules at the aft facing tube inside the HIMIL rather than at the forward facing entrance to the HIMIL. The relative airspeed inside the diffusing HIMIL pylon is considerably slower than outside and flow is laminar.

Motivated by concerns over the potential for cabin air to contaminate the sample stream through outward leaks more forward on the aircraft (Sect. 4.6.3), during ARISTO-2015 we evaluated a new fin HIMIL inlet design consisting of aft-facing 7.9 mm OD, 6.5 mm ID tubes extending 40 cm from the fuselage supported by (and with all but 5 mm contained within) a fin-shaped aerodynamic pylon. ARISTO-2015 was conducted on the NCAR C-130. To evaluate the new inlet design, we switched between the new fin and a standard diffusing type HIMIL at ambient pressures between 400 and 800 hPa, and airspeeds of 110 to 160 m

s$^{-1}$. Comparisons of Medusa flasks taken with the two inlets showed differences of (fin minus diffusing) +11.1 ± 3.4 (standard error) and +3.2 ± 0.9 (standard error) per meg for $\delta(Ar/N_2)$ and APO, respectively. The fin minus diffusing HIMIL APO differences from AO2 were of the same sign and similar magnitude below 5 km, but 2–4 times larger from 5–8 km. Based on the sign of these differences being consistent with greater aft-facing inlet fractionation by the diffusing HIMIL, we decided to use a single fin HIMIL for both AO2 and Medusa on ORCAS and ATom-1.

However, during ORCAS, AO2 measurements between ambient pressures of 200 and 400 hPa and true air speeds > 215 m s$^{-1}$ showed a trough of low $\delta(O_2/N_2)$ values in comparison to Medusa flasks by up to 40 per meg at 300 hPa and occasional 1-min excursions of up to ± 50 per meg. High altitude Medusa flask $\delta(Ar/N_2)$ values from ORCAS did not appear anomalous in comparison to other campaigns (Birner et al., 2020), suggesting the negative features seen in AO2 data were not realistic. This also points to a fractionation effect on a very small scale near the tip of the inlet that may have been more rapidly flushed



by the greater Medusa sampling flow. The full set of ORCAS Medusa results and comparisons to AO2 were not available in time to detect and address this issue before ATom-1, and the same fin HIMIL was again employed.

During ATom-1, the fin HIMIL was located 94 cm above the DC-8 wing and 4.5 m directly behind a 24 cm diameter aerosol collection inlet. On ATom-1 test flights, we observed abrupt decreases of up to 200 per meg in $\delta(O_2/N_2)$ measured by AO2 at
ambient pressures < 300 hPa and true air speeds > 208 m s$^{-1}$ that were coincident with abrupt drops in pressure at our inlet of up to 140 hPa (Fig. S7. The precise mechanism for these large perturbations to the flow at our inlet, whether caused by proximity to the wing or the large aerosol inlet, as well as for the impact on measured $\delta(O_2/N_2)$ at the AO2 inlet is not entirely clear. Investigators sampling at this same location on the DC-8 on previous campaigns anecdotally reported similar pressure effects, and collaborators on ATom sampling 1.8 m behind the aerosol inlet and more forward above the wing observed similar
pressure effects that were dependent on small changes in the distance of their inlet from the fuselage.

During the southbound Pacific flights of ATom-1, we modified our inlet several times in attempts to address the fractionation issue. An attempt to enhance turbulence at the inlet by modifying the inlet edge made the effect more variable and of opposite sign, with excursions up to +200 per meg in $\delta(O_2/N_2)$ at high altitude. Inserting a 3.2 mm OD, 2.2 mm ID tube through and extending 5 cm aft of the existing 7.9 mm inlet tube resulted in larger negative excursions of up to -300 per meg in $\delta(O_2/N_2)$ at
high altitude. Bending this 3.2 mm OD extension to sample 2.5 cm outboard from the pylon and side facing was much worse, and when climbing through ambient pressure of 350 hPa sampling from this side facing inlet we observed abrupt fractionation of $\delta(O_2/N_2)$ by -1400 per meg and fractionation of $CO_2$ by -2 ppm, in the expected ratio for mass dependent fractionation (Fig. S8; Keeling et al., 1998). Because we had 2 inlets connected by a 3-way valve on all these flights, we were able to switch back to our original inlet as soon as we discovered these attempted improvements were not successful. Finally, before the
later northbound Atlantic portion of ATom-1, we found that by bending the trailing 3.2 mm tube 180 degrees into a forward facing inlet, we were able to eliminate the abrupt $\delta(O_2/N_2)$ depletions and had only a relatively consistent pressure dependent offset with Medusa (Fig. 8. Despite a relative improvement in performance with this forward inlet, several times during the remainder of ATom-1 we experienced an obstructed inlet and intermittent flow after flying through liquid water clouds, and thus still prefer aft facing inlets in general.

On ATom-2, we went back to using a diffusing HIMIL, but with a modified design to further reduce the potential for cabin air leaks, and mounted the inlet such that the intake was 18 cm higher and 9 cm further from the fuselage. These changes eliminated the dramatic pressure and AO2 $\delta(O_2/N_2)$ effects seen on ATom-1. Since moving the inlet location and returning to a diffusing HIMIL for ATom 2-4, we have done further speed tests and switching between inlet sizes and orientations inside the HIMIL tube, and do not observe any signs of inlet fractionation in these tests. With AO2 we observed oscillations of up to
$\pm$ 15 per meg during pitch maneuvers done for testing purposes during ATom 2–4. These pitch effects could either be related to unresolved inlet fractionation or to the dynamic inlet pressure and humidity effects described below (Sect. 4.4).

While Medusa flask samples in general show less evidence of fractionation than AO2 measurements on ORCAS and ATom-1, we do find evidence of systematic offsets in Medusa $\delta(Ar/N_2)$ across all campaigns between the various inlets and configurations used. Comparing the mean boundary layer Medusa flask $\delta(Ar/N_2)$ values from each campaign (Fig. S4) shows an
approximate range of 0 to 30 per meg, with the mean of HIPPO campaigns (3.7 $\pm$ 3.0 per meg, n=5) lower than ATom 2-4





(12.9 $\pm$ 2.2 per meg, n=3), and both of these lower than ORCAS and ATom-1 using the fin HIMIL (28.0 and 29.2 per meg respectively) (Table S3).

The indication from ARISTO-2015 on the C-130 that the diffusing HIMIL fractionated more than the fin HIMIL despite the fin HIMIL clearly fractionating more on the GV and DC-8, remains unresolved. Possible explanations include the differing air

speeds of these aircraft or different orientations of the inlets to the relative wind flow. A systematic study of inlet fractionation in a high speed wind tunnel would be a valuable contribution to the airborne greenhouse and related gas measurement field. That the HIMIL design works as well as it does for $\delta(O_2/N_2)$ and $\delta(Ar/N_2)$ sampling is likely attributable to its heritage as an aerosol inlet. It was designed to reduce the well-known tendency of aircraft inlets to differentially sample heavy and light aerosol particles (e.g., Belyaev and Levin, 1974), a similar effect to our observed separation of heavy versus light molecules.

While a forward facing inlet also appeared to reduce fractionation effects on the UND Citation II during COBRA, the forward inlet on the second half of ATom-1 showed modest positive fractionation, and forward inlets also come with the increased risk of ingesting liquid water, insects, or other debris.

### 4.2.1  Adjustments and filtering for inlet fractionation

For Medusa an adjustment for inlet fractionation in the troposphere is mostly included in the calculation of $\delta(O_2/N_2)^*$ described

above (Sect. 4.1.1), since we can not distinguish deviations in $\delta(Ar/N_2)$ caused by thermal or pressure-gradient fractionation. The expected mass-dependent relationship between $\delta(O_2/N_2)$ and $\delta(Ar/N_2)$ for the inlet fractionation effect proposed by Steinbach (2010) is 3.0. However, the error in using the thermal 3.77 ratio for the inlet effect is only 1.5 per meg in $\delta(O_2/N_2)$ over a typical HIPPO 10 km vertical gradient. For stratospheric Medusa samples, we adjust for an estimated inlet fractionation effect based on a linear extrapolation of the tropospheric $\delta(Ar/N_2)$ fit to pressure (Bent, 2014). This adjustment is done before the

fit between $\delta(Ar/N_2)$ and $N_2O$ in generating the stratospheric portion of the reference $\delta(Ar/N_2)$ profile (Sect. 4.1.1). Although the $\delta(Ar/N_2)$ vertical gradients and scatter on ATom 2–4 suggest that thermal and inlet fractionation were minor effects (Fig. S4, S5, S6), we calculate Medusa $\delta(O_2/N_2)^*$ for consistency with other campaigns. We have not implemented the $\delta(O_2/N_2)^*$ calculation for COBRA Medusa samples.

For AO2 on ORCAS, we filter out data between ambient pressures of 200 and 400 hPa when the airspeed was greater than

215 m s$^{-1}$, based on up to 40 per meg negative deviations with respect to Medusa under these conditions. This filter removed 19 % of the available data from ORCAS, but none of the lower altitude data of particular interest for this Southern Ocean gas exchange study. For the first half of ATom-1 we filter a subset of the data on research flights 2, 4, and 5 collected with unsuccessful inlet modifications. For all of the first half of ATom-1 flights using the aft facing 7.9 mm inlet, we filter data when the airspeed was greater than 208 m s$^{-1}$, based on the up to 200 per meg negative deviations with respect to Medusa under

these conditions. These filters removed 27 % of the available data from ATom-1. For the second northbound half of ATom-1, using the forward facing 3.2 mm inlet, we apply flight-specific linear pressure dependent corrections to AO2 $\delta(O_2/N_2)$ values based on differences to Medusa, that are on average 0 per meg at 1000 hPa and -27 per meg at 200 hPa (Fig. 8). For this correction, we compare to Medusa $\delta(O_2/N_2)^*$ values. The $\delta(O_2/N_2)^*$ calculation largely addresses scatter, systematic offsets, and vertical gradients owing to thermal and pressure fractionation effects. In some cases it may be desirable to correct only for





some of these effects. For example one could apply a single offset for each campaign with a constant vertical gradient, which would largely address the inlet fractionation offsets while still leaving the original scatter and avoid contributions from real $\delta(Ar/N_2)$ variations.

### 4.3    AO2 regulator flushing and tubing conditioning

The two-stage cylinder pressure regulators we use are commonly used for high precision laboratory $\delta(O_2/N_2)$ and $CO_2$ measurements, but have elastomer seals and are recognized to require flushing before producing stable readings. The volume of air required for flushing depends on the length of time the regulators have been stagnant, but can be several liters or more if it has been several days. We start cycling through calibration gases an hour or more before takeoff. The high and low span gases typically get purged and analyzed 6 times during preflight, using a total of 3 l of gas. During this warm up period, we often

observe increasing $\delta(O_2/N_2)$ and $CO_2$ readings for the calibration gases. We speculate that these trends result from drying of either the regulator seals or tubing surfaces, causing $O_2$ and $CO_2$ to adsorb or absorb in place of the removed $H_2O$. Both $H_2O$ and $O_2$ adsorb to stainless steel, but $H_2O$ adsorbs preferentially and prevents $O_2$ adsorption (Buckley, 1968). $N_2$ does not adsorb to steel at ambient temperatures (Armbruster and Austin, 1944). Such an effect would produce negative biases in both gases and be most pronounced initially, when the rate of drying was greatest, and decrease as the drying proceeded and slowed.

Negative biases for calibration gases would result in positive biases for ambient air measurements and thus this effect could lead to AO2 $\delta(O_2/N_2)$ biases that trended downwards during flight. Conversely, if the AO2 trap were inadequately drying sample air the tubing downstream of the cal-sample selection valve could be wetting while measuring ambient air and drying while measuring calibration gas. This might bias the ambient air measurements high for $\delta(O_2/N_2)$ with more complex time of flight dependencies, depending on how dry this tubing was initially. A feature of the AO2-Medusa comparisons during HIPPO was

$\delta(O_2/N_2)$ differences that trended downwards during flight (Fig. S9), with magnitudes ranging from -2 to -5 per meg hour$^{-1}$. During HIPPO, we ran calibration gases starting 45 minutes instead of 1 hour before take-off, and our trap swapping procedures may have allowed wet ambient air into the system on maintenance days. Furthermore, our first and second stage traps were not drying efficiently (see Sect. 4.5). This time of flight dependency has been improved since ORCAS (Fig. S9), with dry purge gas flushing of the inlet system on maintenance days, more preflight regulator flushing, trap swapping procedures that prevent

ambient air introduction, and better drying efficiency. Notably, the time-of-flight AO2 $\delta(O_2/N_2)$ dependencies during HIPPO, or during other campaigns, do not appear to be related to cylinder drift. We measure and log temperature at 6 points inside the AO2 cylinder box and do not find correlations across flights with trends in either temperature or temperature gradients in the box.

### 4.3.1    Adjustments for time-of-flight dependent biases

Because the Medusa flow rate is 15–27 times greater than for AO2, and we find no other evidence in Medusa flask measurements of time-of-flight dependencies, we attribute time-of-flight dependencies in the AO2-Medusa difference to biases in AO2 and a combination of the calibration cylinder regulator and line drying and sample line wetting described above. Therefore, we adjust the AO2 $\delta(O_2/N_2)$ data using flight-specific linear time-dependent fits to the AO2-Medusa differences for each flight, as





summarized for campaign means in Fig. S9. These fits are made to differences between AO2 $\delta(O_2/N_2)$ and Medusa $\delta(O_2/N_2)^*$. The mean impact of these linear time-of-flight adjustments for all flights is -1.7 $\pm$ 1.8 per meg hr$^{-1}$ (1 $\sigma$). For ATom 2–4 the mean time-of-flight adjustment is -0.9 $\pm$ 0.9 per meg hr$^{-1}$ These adjustments remove both the apparent trend in AO2 data over a flight, and also the mean difference between AO2 and Medusa for reasons described below (Sect. 4.5).

## 4.4 AO2 differences between ascent and descent

There are large and dynamic differences in inlet and instrument condition when the plane is ascending versus when it is descending. These differences include: the angle of attack at the inlet; the relative air speed at the inlet; the sign of change of inlet pressure, temperature, and humidity; the angle of the instrument with respect to gravity; and the sign of change in cabin pressure. To assess the potential for bias between ascent and descent, it is useful to compare measurements from adjacent profiles.

During ACME-07 and the first half of START-08, the AO2 inlet included long sections of non-pressure-controlled 6.4 mm OD, 4.3 mm ID ethylene copolymer lined tubing, 2 m and 7 m respectively. We found that as a result the AO2 measurements were biased in proportion to the rate of change of the pressure in this tubing, with $\delta(O_2/N_2)$ biased low during descents and high during ascents by up to $\pm$ 75 per meg during ACME-07 and $\pm$ 200 per meg during the first half of START-08. We attribute this effect to the preferential absorption and desorption of either $O_2$ relative to $N_2$ or $H_2O$ relative to $O_2$ into and out of the ethylene copolymer lining. Between the first and second phases of START-08 (before Research Flight 13) we replaced this tubing with 2.2 mm ID stainless steel tubing and moved the inlet pressure control valve from the AO2 rack to immediately inside the aircraft, which eliminated these large pressure dependent biases. Medusa also sampled from 6.4 mm OD, 4.3 mm ID ethylene copolymer lined tubing during START-08, but with 15–27 times the flow rate of AO2, and we did not find any clear evidence of Medusa sampling biases from inlet pressure changes. To further reduce the likelihood of surface interactions, for the AO2 inlet line we switched to electropolished 2.2 mm ID stainless steel tubing before HIPPO-1 and electropolished Sulfinert treated 2.2 mm ID stainless steel tubing before HIPPO-4. For the Medusa inlet line we switched to 4.6 mm ID electropolished stainless steel tubing and moved the inlet pressure controller to immediately inside the fuselage before HIPPO-2.

Despite greatly improved performance after moving the AO2 pressure control point upstream and switching to stainless steel tubing, followed by further reductions in surface roughness, we still see small differences between $\delta(O_2/N_2)$ on ascents versus descents for AO2. As shown in Fig. S10 for HIPPO, ORCAS, and ATom, the magnitude of these differences are generally greatest at altitude and can be as large as -10 per meg on average, for the first half of ATom-1 and for ATom-2. Comparisons to Medusa and with level legs at altitude suggest these biases were symmetric, with the end of the ascents biased low by 5 per meg and the start of the descents biased high by 5 per meg. HIPPO-1 showed ascent minus descent differences of opposite sign, with a peak of 5 per meg ($\pm$ 2.5 per meg) at mid altitudes (Fig. S10). HIPPO-2 through HIPPO-5 had campaign average ascent minus descent differences close to zero but a tendency towards negative differences at altitude consistent in sign with later campaigns. ORCAS ascents and descents were also similar at pressures greater than 600 hPa but diverged by up to 7 per meg between 400 and 600 hPa. The first and second halves of ATom-1 and ATom-2 showed the largest ascent minus descent





differences, which peaked between 650 and 450 hPa at -9 to -14 per meg. Finally, the ascent minus descent differences on
ATom-3 and 4 were very consistent with a peak at max altitude of -5 per meg.

Calibration gas measurements do not show this behavior so we can rule out cabin pressure or pitch effects on the AO2
instrument. Of the external changing parameters, those with greater ascent versus descent differences at altitude include angle
of attack and the relative rate of change in water vapor concentration, pointing to either inlet fractionation or surface effects.
Also, the effect appears to vary from flight to flight and within a flight, which could result from variable flight conditions
or tubing surface conditions. However, the ascent minus descent differences in angle of attack on ATom-4 were a factor of 2
greater than on previous campaigns, with no noticeable difference on the $\delta(O_2/N_2)$ sensitivity. Finally, the effect did not reverse
sign between the aft and forward inlets used in ATom-1, which might be expected for an inlet fractionation effect.

The ascent minus descent differences since HIPPO-1 are opposite in sign to the earlier ethylene copolymer lined tubing
effect, but are consistent in sign with the slower tubing and regulator drying effect described above (Sect. 4.3). We suspect
a similar competition between $H_2O$ and $O_2$ on tubing surfaces is responsible (Buckley, 1968). In this scenario, on ascent as
pressure and humidity decrease in the AO2 inlet upstream of the control valve, and humidity decreases in the inlet and tubing
upstream of the cryotrap, there would be less competition for $O_2$ adsorption leading to a net loss of $O_2$ from the sample air
to the tubing surfaces. Conversely as humidity and pressure increase on descent more $H_2O$ would adsorb leading to a net
desorption of $O_2$ from the tubing surfaces to the sample air. The effect is largest at altitude despite the absolute rate of pressure
and humidity changes being largest at low altitude, but this could be explained by a saturation of $H_2O$ adsorption at relatively
low humidities. The colder inlet temperatures at altitude could also play a role.

### 4.4.1 Adjustments for ascent versus descent differences

For ACME-07 and the first half of START-08 we have adjusted the AO2 data using linear fits between APO and a smoothed
representation of the time rate of change in pressure at the inlet, optimized by adjusting the smoothing window. This adjustment
is zero when inlet pressure is not changing and at other times is negative or positive with a magnitude determined by the
optimized fit. Although the empirical correlations for these adjustments are reasonably good ($r^2$ values from 0.5 to 0.9), we
suggest caution in detailed interpretations of the individual AO2 $\delta(O_2/N_2)$ profiles from these flights, as significant biases
may remain. However, by either looking separately at results from ascents versus descents or averaging data from ascents and
descents, the impact of these biases on particular results can be identified or largely removed.

For the smaller effect seen on more recent campaigns (HIPPO, ORCAS, and ATom), we derive an empirical correction
based on comparisons between subsequent profiles. For each profile, we subtract a combination of the prior and following
profile, interpolating by pressure, then fit these differences with a linear relationship to pressure, excluding profiles with > 40
% missing data. We then compute a 4-profile running mean of the bias versus pressure slopes, to allow for trends within a flight
while avoiding real atmospheric differences on a single profile from having too much influence. Finally, we interpolate these
smoothed slopes to all times in the flight and use them to calculate a correction to the flight data depending on whether the
plane was climbing or descending, and at what pressure altitude. While the most noticeable impact of this correction is better
visualization of upper-tropospheric patterns in $\delta(O_2/N_2)$ in cross-section plots (e.g. Fig. 11), it will also improve results based





on vertical gradients in individual profiles. Analyses that average multiple profiles together, such as the mean vertical gradient over a flight or region, are largely unaffected, as the corrections are balanced from one profile to the next.

## 4.5 AO2 water and hydrocarbon effects

During the HIPPO campaigns we used simple 40 cm long 6.4 mm OD u-shaped tubes for the second stage sample air and working tank cryotraps, and a narrower 9.5 mm ID first stage sample air cryotrap. Over the course of the 5 HIPPO campaigns, the differences between AO2 and Medusa $\delta(O_2/N_2)$ measurements became steadily more negative, to reach a minimum of approximately -80 per meg during HIPPO-4 and HIPPO-5 (Figs. 8 and S9), despite significant efforts between each campaign to diagnose and address these offsets. We also observed differences in the slope of subsequent working tank and span gas jogs during these campaigns, on the order of 30 per meg s$^{-1}$ (Fig. 3). Laboratory tests after HIPPO-5 finally confirmed that the cryotraps were not adequately drying sample gas before it entered the VUV cell. Although measurements of ppm-level H$_2$O at our sample flow rate are challenging, our best estimate using a laboratory dew-point hygrometer is that during HIPPO-5 the second stage trap outlet had on the order of 15 ppm of H$_2$O when sampling outside air.

We can exlude a direct VUV absorption effect from this water because the biases were in the opposite direction from that expected for additional absorption. Less water would likely have exited the traps during calibration periods, but trap and tubing surfaces would have contributed water to the dry calibration gas resulting in transient responses in H$_2$O over calibration sequences, and we did observe transient $\delta(O_2/N_2)$ changes for several minutes after each cal-sample and sample-cal switch on these campaigns. Alternate wetting and drying of surfaces downstream of the cal / sample selection manifold might be expected to lead to O$_2$ adsorption and desorption in the other direction (Sect. 4.3), however this also would have resulted in AO2 biases with the opposite sign of those we observed. Nonetheless, replacing the u-tube traps with longer and smaller diameter coiled traps, and increasing the diameter of the first stage trap, eliminated the transients and greatly reduced the AO2-Medusa differences in ARISTO-2015 and subsequent campaigns (Figs. 8 and S9), despite our not having a good explanation for the cause of the bias at the time. These changes also eliminated the working tank versus span jog slope differences.

Then, between the ATom-3 and ATom-4 campaigns, two discoveries led us to hypothesize that the biases during HIPPO were likely a result of photochemical dissociation of H$_2$O in the detector cell followed by radical interactions with optical surfaces. The first was that when using AO2 working tank gas from a commercial vendor that been scrubbed of all hydrocarbons in comparison to compressed natural sample air, the differences in the slopes of subsequent working tank and span jogs was around 20 per meg s$^{-1}$, in the same direction as with wet sample gas, in comparison to zero when both gases had ambient CH$_4$ concentrations. The second came while conducting tests in the laboratory, sampling air from large polyethylene barrels used as integrating volumes. Switching to sampling barrel air led to large increases in the working tank versus span jog slope differences. Then, after switching back to sampling inlet line purge cylinder air, the measurements were biased low and both the jog slope differences and the biases persisted for at least 2 hours. Either replacing the sample trap or warming it with a heat gun under vacuum for several minutes and then rechilling it eliminated the problem.

Notably, the problem of inadequate drying on HIPPO, the difference between gas with and without hydrocarbons, and the polyethylene barrel problem all manifested themselves similarly in terms of the direction of working tank versus span jog





slope differences and $\delta(O_2/N_2)$ biases. Specifically, when water or excess hydrocarbons are present in the span gas relative to working tank gas, the slope of the VUV signal during span jogs is positive, indicating decreasing $O_2$ or increasing light. The signal slope is opposite during working tank jogs, and the measured $\delta(O_2/N_2)$ values for the span gas are biased low. VUV absorbtion by $H_2O$ or $CH_4$ is too weak (Stephens, 1999) and of the wrong sign to explain these effects. Also, the photochemical

production of another absorbing species can not explain the trends over several seconds as the residence time of the air in the sample cell is on the order of 0.01 s. However, photochemical processing of $H_2O$ and hydrocarbons in the intense VUV light may result in a "cleaning" effect on the lamp and detector optics, via surface reactions with OH or other radicals. Such an effect would be consistent with the increasing signal over several seconds during span jogs leading to the appearance of less $O_2$ in the cell. We now avoid using commercially sourced gases lacking ambient $CH_4$. Also, since HIPPO we have ensured that

the traps are drying the air sufficiently, and have adjusted our procedures to avoid introducing wet ambient air into the system when swapping traps between flights.

### 4.5.1 Adjustments for inadequate drying of air

For HIPPO 1-5, we make a constant adjustment to AO2 $\delta(O_2/N_2)$ for each flight based on the comparison to Medusa (Figs. 8 and S9). These adjustments are in combination with the time-of-flight slope adjustments (Sect. 4.3.1) and thus have the effect

of adjusting AO2 by the average offset for each flight. These comparisons are made to Medusa $\delta(O_2/N_2)^*$.

### 4.6 Additional measurement considerations

In addition to the challenges described above, in this section we discuss several other aspects of high precision airborne $O_2$ measurements that require careful attention.

### 4.6.1 Propagation of AO2 calibration scales

A critical requirement for the AO2 measurements is the propagation of primary calibration scales for $\delta(O_2/N_2)$ from Scripps and $CO_2$ from both Scripps and NOAA/GML. Our laboratory primary cylinder suite consists of six 50 l aluminum cylinders originally filled, adjusted, and calibrated at Scripps in 2005 and calibrated at NOAA in 2006. These cylinders have return to Scripps and NOAA every 5 years since for reanalysis to maintain our links to the Scripps $O_2$ Program $O_2$ and $CO_2$ scales and the WMO $CO_2$ scale. Our internal laboratory scales are then defined by linear interpolation of these external measurements.

Over 15 years our primaries have varied by $< 5$ per meg $\delta(O_2/N_2)$ and $< 0.05$ ppm $CO_2$. We propagate these scales to secondary cylinders annually and then to our flight cylinders before and after each campaign, and we show these results in Fig. S11.

Stability for $\delta(O_2/N_2)$ and $CO_2$ is often an issue in larger high pressure cylinders (Langenfelds, 2002; Keeling et al., 2007) and even more of a concern for smaller cylinders which could amplify fractionation effects. We initially valved our cylinders using viton o-rings but found drifts on the order of -100 per meg over 1 year (Fig. S11). Starting with HIPPO-2, we used silver

coated c-rings (Technetics Helicoflex) for all but our working tank and inlet line purge cylinders, and for all cylinders after HIPPO-5. Cylinders with these silver seals are generally very stable for both $\delta(O_2/N_2)$ and $CO_2$, with positive drifts less than





5 per meg over 1 year, with a few outliers showing drifts up to 60 per meg in the first 4 months (Fig. S11). The cause of these more recent outliers is unclear, but may be related to inadequate drying or a faulty regulator. We now measure the humidity in each cylinder and our filling procedures routinely achieve humidities of less than 1 ppm $H_2O$. We select our flight span and long term reference cylinders from those showing the best stability in lab. For all campaigns, we measure the field cylinders

for several weeks in lab immediately before and after the deployment, and assume a linear drift in time between the average prior and post campaign laboratory determinations.

### 4.6.2   Cabin temperature and pressure effects

It is also possible for temperature variations to cause separation of gases within a cylinder and thus affect $\delta(O_2/N_2)$ values in the gas exiting the head valve (Keeling et al., 2004, 2007). We mount our cylinders horizontally in an insulated enclosure in an

attempt to minimize these effects. Also, we use dip tubes to withdraw air from the middle of the cylinder, following practices established for laboratory cylinders by Keeling et al. (2007). However, the temperature changes on research aircraft can be very large and it is not practical to isolate the cylinders from more than short term fluctuations. To support detection and diagnosis of temperature effects on our cylinders, we measure temperature at 6 locations within the AO2 cylinder box, distributed to detect temperature gradients in 3 dimensions. We have compared gradients and trends in these temperatures to the calibration gas

measurements and to differences between AO2 and Medusa flask measurements for all campaigns. We are unable to identify any relationships attributable to thermally induced fractionation of the delivered cylinder air.

We also measure temperatures at various locations in the AO2 instrument and pump boxes, as well as cabin pressure. The voltage output of the AO2 detector is tightly linked to the temperature of the lamp and detector housing, likely reflecting changes in lamp output, cell pressure and air density, and amplifier gain. However the effect of the temperature dependent

trends in raw detector voltage are generally imperceptible on the amplitude of the square wave in voltage from switching between span and working tank air.

To monitor potential cabin pressure effects, we measure the ambient pressure inside the AO2 pump box and look for correlations with reference cylinder measurements and other system diagnostics. For ORCAS, we moved the sapphire window in the AO2 detector to the lamp side rather than detector side of the cell, which resulted in the lamp being secured only by

a teflon clamp rather than than also being pulled flush to the cell by the low cell pressure. As a result, when cabin pressure changed in a climb or descent, the raw detector voltage smoothly oscillated by up to 0.02 V with an approximate 2 min period, possibly related to resonant heating and cooling of the teflon clamp or magnet wire coiled around the lamp resulting in subtle movements in the lamp itself. These oscillations resulted in increased noise in the square wave signal, which we were able to remove by applying a loess fit (Cleveland and Devlin, 1988) to the working tank measurements, rather than straight interpola-

tion, in calculating the amplitude of the square wave. We eliminated these oscillations before ATom-2 by returning the sapphire window to the detector side of the cell.



### 4.6.3 Cabin air leaks

The combination of human respiration, dry ice sublimation, and liquid nitrogen evaporation within the typical research aircraft cabin can lead to highly perturbed $CO_2$ concentrations and $O_2/N_2$ ratios in the cabin air. On several flights on both the GV and DC-8, we used AO2 to measure $CO_2$ and $\delta(O_2/N_2)$ of cabin air, and saw typical values enhanced by 250 ppm $CO_2$ and depleted by 500 per meg $\delta(O_2/N_2)$. Even small contributions of cabin air to our sample gas, either through direct leaks in our inlet plumbing, or via fuselage vents or leaks upstream of our inlet, could potentially affect our measurements (e.g., Vay et al., 2003). In addition, a leak from the cabin to the sample stream through a small orifice could further deplete $\delta(O_2/N_2)$ by contributing air fractionated through the process of Knudsen diffusion (Keeling et al., 1998).

During maintenance days and in preflight we routinely conduct vacuum and pressure leak checks on all AO2 and Medusa plumbing to carefully monitor for and detect any system leaks. In addition, several times per flight while at high altitude, we bathe our low pressure inlet fittings with pure $CO_2$ from a bottle of dry ice, and monitor the AO2 and Medusa $CO_2$ signals for any spikes that would indicate leaks. These procedures have proven sufficient for eliminating leaks in the instruments and the portion of the sample tubing which is inside the cabin.

However, pressurized aircraft are not airtight, and for example, potential sources of cabin air upstream of our inlet on the GV include the cabin dump valve, a separate cockpit air conditioning vent both on the lower right of the forward fuselage, a large gasket door seal on the forward left side, the nose compartment, and leaks from other instrument inlets. The DC-8 has similar concerns, including a forward lavatory vent on the left side of the fuselage. On any research aircraft, atmospheric sampling inlets must extend beyond the boundary layer of the fuselage to sample uncontaminated air. For the GV and the DC-8 the aircraft boundary layer grows from the front of the aircraft at approximately 1 and 1.2 cm per m, resulting in predicted depths of 23.5 and 11.0 cm for our AO2 inlet locations on HIPPO and ATom, respectively. During test flights on the GV during the 2005 Progressive Science campaign, pressure was measured at several locations and a range of distances from the fuselage to empirically determine the boundary layer depth. These tests indicated that the aircraft boundary layer extended to approximately 21 cm at the locations of the AO2 and Medusa inlets during HIPPO and ORCAS. Since our inlet extended 30.5 cm out from the aircraft, we expect this length was sufficient to sample undisturbed air.

Nonetheless, the growing negative AO2-Medusa $\delta(O_2/N_2)$ offsets we found over the course of HIPPO (see Sect. 4.5, Figs. 8 and S9, and Table S3), led us to vigorously investigate potential cabin air leaks. In particular, we were concerned about potential leaks in the HIMIL itself. These might occur through the many o-rings, gaskets, or screw holes that allow for heating the inlet for other instrument applications and passage of the tubes through the pylon, or through sheathed tubing used by another instrument sharing a HIMIL, as was the case for AO2 in HIPPO. However, laboratory test on the HIMIL and further tests with pure $CO_2$ in flight failed to confirm our suspicions, and we also did not find correlations between the offsets and cabin pressure, ambient pressure, or their difference, that might suggest a leak. As described in Sect. 4.5, subsequent laboratory tests confirmed that inadequate drying and not cabin air leaks was the primarily cause.





### 4.6.4 AO2 pressure and flow control

Many of the challenges described above, including inlet fractionation and regulator and tubing conditioning, could be mitigated by higher flow rates. However, we have not been able increase the flow rate without also increasing the short term instrument noise. In laboratory tests, increasing the flow rate by increasing the upstream reference volume pressure or swapping in a larger

sapphire orifice has led to 2-4 times greater noise for the 5 second measurement, and typically a smaller uncorrelated increase in the noise of the pressure signal from the downstream differential pressure transducer. Also, on several flights after increasing the flow rate slightly while on the ground, once in flight the detector signal varied rapidly by the equivalent of 100 per meg before stabilizing after flow was reduced, suggesting an instability in the pressure control. Prior to HIPPO-1, we used a 5 cm by 0.25 mm ID capillary in place of the sapphire orifice, and before ATom-3, we removed a 10 μm by 6.4 mm diameter screen

that was acting as a damper between the cell and the downstream pressure transducer. Neither of these changes dramatically changed the measurement precision or its sensitivity to increased flow. Pressure and flow control at the $10^{-6}$ relative level depends on many factors, including flow restrictions, pressure transducers, fast-response proportional valves, and the tuning of the feedback control circuitry. Ongoing laboratory work will continue to explore improved pressure and flow control at higher flow rates.

### 15    4.6.5 AO2 motion sensitivity

Sensitivity of the $O_2$ measurement to motion can arise from movement of the lamp and ballast coils, movement of components within the extended RF field of the lamp, and from acceleration effects on the proportional solenoid valves used for pressure control. We secure the lamp coils with a teflon clamp and glue between coils, which appears to eliminate this potential source of noise. We also correct for measured deviations in pressure control, which can be large during turbulence, and this correction

is effective at reducing the solenoid valve contribution. However, RF coupling has been more challenging to address.

During the first test flights of the VUV sensor during IDEAS-1 in 2002, we discovered the RF field was escaping from the lamp box, and movement of the mounting plate for the lamp and detector box relative to the rest of the rack led to large motion effects. This was largely fixed by improving the shielding of the lamp box. However, throughout HIPPO a small amount of motion sensitivity persisted, with short term noise during a typical boundary-layer leg increasing by a factor of 2-3, and more

so under moderate turbulence. Then during ARISTO-2015, after we found that adding vibration isolators to the rack made the motion sensitivity worse, we discovered that by better securing the wires and cables inside the lamp box, we were able to eliminate most of the remaining motion sensitivity. As flown during ORCAS and the ATom campaigns, short-term noise during boundary-layer legs was typically indistinguishable from other portions of flight, but occasionally in moderate turbulence it was approximately a factor of 2 greater.

### 30    4.7    Independent performance checks

To assess the propagation of laboratory calibrations to the in situ AO2 measurements, and other aspects of the instrument stability, we measure the long-term surveillance cylinder multiple times during preflight and approximately every 100 minutes





during flight. Fig. S12 shows $\delta(O_2/N_2)$ differences between these measurements and our laboratory determinations for these cylinders. The campaign mean offsets are shown in each panel and in Table S3. Across all campaigns, these differences have a mean of -1.9 ± 7.8 per meg (1 $\sigma$, n = 599). From ARISTO2015 on, the mean offset is -0.7 ± 4.1 per meg (1 $\sigma$, n =349), and for just ATom-3 and 4 the offset is -0.8 ± 1.9 per meg (1 $\sigma$, n = 157). However, during HIPPO, the long-term surveillance

measurements showed systematic biases of up to ± 10 per meg, and -20 per meg for the first half of HIPPO-2.

Negative biases on HIPPO-4 and 5, and on the first half of HIPPO-2, can be attributed to transient slopes during the long-term surveillance measurement itself, owing to inadequate flushing of the long-term surveillance cylinder regulator and lines before measurement. Conversely, positive biases on the second half of HIPPO-2 and on HIPPO-3 likely result from a greater impact of inadequate flushing of the high and low span cylinders which precede the long-term surveillance cylinder measurement. These

offsets are generally smaller than those we attribute to inadequate regulator flushing and tubing drying during the HIPPO campaigns (Sects. 4.3 and 4.5) and are accounted for by the adjustments described in Sects. 4.3.1 and 4.5.1. Overall, these long-term surveillance results demonstrate that errors in our propagation of calibration scales from the laboratory to field measurements are now a relatively small component of overall AO2 $\delta(O_2/N_2)$ measurement uncertainty.

We have assessed the magnitude of any overall biases in Medusa by comparing our airborne observations from the HIPPO,

ORCAS, and ATom campaigns to biweekly station flask samples collected and analyzed by the Scripps $O_2$ Program (Keeling and Manning, 2014). We use samples from all 10 stations in the Scripps $O_2$ Program network. Because we only occasionally flew close to stations, and only rarely on the actual day of a flask collection, we must use relatively loose coincidence criteria. We select any Medusa flasks that occur within 1000 km horizontally and 1000 m vertically of a sampling station, and within 10 days of a station flask collection. Next we interpolate the station measurements in time to match the date and time when

the aircraft was nearest. We then take the median of the selected Medusa measurements for each match and subtract the time-interpolated station measurements. The average results of these comparisons are tabulated for all campaigns in Table S3. On the basis of APO*, the mean offset between Medusa and station measurements for all campaigns was 0.2 ± 8.2 per meg (1 $\sigma$, n=86). This comparison is to measurements on station flasks using the same mass-spectrometer as for Medusa flasks. Individual campaign means vary from -4.9 to 5.3 per meg (average n=9) with a standard deviation of ± 3.3 per meg (Table S3). This range

in campaign means is as expected for random sampling of the full population, and suggests a relatively consistent relationship over time.

With confidence in the overall quality of the Medusa flask measurements, we then evaluate AO2 measurements by comparison to coincident Medusa flasks. Figs. 8 and S9 show these differences and Sect. 4 discusses adjustments made to account for the large offsets primarily seen during the HIPPO campaigns. Since resolving the inadequate drying issues present in HIPPO,

the 6-campaign mean unadjusted AO2-Medusa offset is -0.3 ± 7.2 (1-sigma, n=1361). Averaged over individual campaigns, the 6 campaign mean offsets since HIPPO range from -4.5 to 5.2 per meg (Table S3).



## 5 Measurement examples

To highlight the performance of AO2 and Medusa and their scientific potential, we present a limited set of examples from past campaigns. These include several vertical profiles, a collection of source-specific correlations between $\delta(O_2/N_2)$ and $CO_2$, and global altitude-latitude cross sections for $CO_2$, $\delta(O_2/N_2)$, and APO.

### 5.1 Vertical profiles

Fig. 9a and b show a vertical profile measured by AO2 over an agriculturally-dominated region during early summer. As indicated by potential temperature, there was a well-mixed boundary layer to approximately 2 km and the tropopause was at approximately 12 km. On this late-afternoon profile, the boundary layer showed an approximate decrease of 6 ppm $CO_2$ and a well-correlated increase of approximately 30 per meg $\delta(O_2/N_2)$ relative to air immediately aloft. The average molar ratio for the variations below 4 km is close to 1 mole $O_2$:mole $CO_2$ (Fig. 10d) and the APO profile is nearly flat (Fig. 9b), indicating a dominant influence of regional terrestrial photosynthesis in producing these signals (e.g. Stephens et al., 2007a; Battle et al., 2019). The overall gradients through the troposphere suggest the seasonal late-spring Northern Hemisphere $CO_2$ maxima and $\delta(O_2/N_2)$ minima were eroding more slowly in the upper than lower troposphere. The jump to lower $CO_2$ and greater $\delta(O_2/N_2)$ values in the lower stratosphere on this profile results from the relative isolation of this air from both the previous winter's Northern Hemisphere seasonal signals and longer term global trends.

Fig. 9c and d show a vertical profile measured by AO2 and Medusa over the Southern Ocean during early spring. In this case, the $CO_2$ profile is nearly flat but the $\delta(O_2/N_2)$ and APO profiles show a strong depletion in the lower 3 km and greater negative excursions within 500 m of the ocean. These signals are consistent with uptake of $O_2$ by the ocean as a result of ventilation of $O_2$-depleted waters and cooling of surface waters over winter. The stratospheric $\delta(O_2/N_2)$ deviation for this profile is the same sign as that in Fig. 9a reflecting both the trend and the tropospheric winter influence. The stratospheric $CO_2$ signal is more muted owing to the small $CO_2$ seasonal cycle at high southern latitudes.

### 5.2 $O_2$ versus $CO_2$ correlations

In addition to natural terrestrial and ocean exchange signals, AO2 and Medusa have often sampled polluted air. Because various fossil-fuel types and terrestrial and oceanic exchanges have distinct $O_2$:$CO_2$ signatures (Keeling, 1988; Steinbach et al., 2011), the molar ratios observed for these events can provide a means of source identification. Fig. 10 shows examples of 3 such events along with the agriculturally influenced profile presented in Fig. 9a. The combustion of fossil fuel exchanges more $O_2$ per molecule of $CO_2$ than terrestrial photosynthesis, because the carbon is more reduced. The ratios observed for a natural gas plant, a city, and a coal plant shown in Fig. 10 are close to those expected for methane, liquid fuels, and coal (-1.89, -1.34, -1.16 observed versus -2.00, -1.43, and -1.15 expected, respectively Keeling, 1988).





## 5.3 Altitude-latitude cross sections

Each month-long HIPPO and ATom campaign included flights north of Alaska to around 87°N, transecting the Pacific southwards to New Zealand, south to around 67°S, and returning north again either via the Pacific (HIPPO) or Atlantic (ATom) basin (Fig. S1). During each HIPPO and ATom flight, the aircraft profiled continuously between a near-surface altitude of 150–300 m and a maximum altitude of 9–14 km. Fig. 11 shows interpolated AO2 altitude-latitude cross sections overlain with Medusa observations for the southbound Pacific basin portion of ATom-4 in April-May of 2018. The $CO_2$ cross-section shows concentrations elevated by over 5 ppm throughout the entire northern extratropical troposphere, with enhancements as high as 8 ppm north of 60°N. This reflects the seasonal accumulation of northern extratropical $CO_2$ emissions over winter from a combination of net terrestrial respiration and fossil fuel burning. The color scales in Fig. 11 are set to be equivalent on a molar basis and show larger northern extratropical depletions in $O_2$ as a result of the greater than 1 oxidation ratio for fossil fuel burning and the additional ocean uptake of $O_2$ resulting from both ventilation of northern ocean waters with accumulated respiration signatures and the cooling of surface waters.

Conversely, at southern high latitudes, ocean heating and net marine productivity lead to $O_2$ emissions over the austral summer which we observed as a strong accumulated $O_2$ signal throughout the southern extratropical tropopause. Given the relative lack of land plants and industrial emissions at high southern latitudes, the observed southern hemisphere $CO_2$ field was comparatively flat. APO effectively masks out terrestrial influences, and suggests that the interhemispheric gradient in $\delta(O_2/N_2)$ at this time of year is approximately half owing to air-sea fluxes. These flights also intercepted stratospheric air poleward of 60°N and less than 300 hPa, and in an isolated intrusion at 33°N and 300 hPa, with correspondingly high $O_3$ and other stratospheric tracers (not shown).

All 5 HIPPO and 4 ATom campaigns with the exception of HIPPO-1 returned to the Arctic at the end of their northbound transects. Thus, we have collected 17 complete global altitude-latitude transects such as those shown in Fig. 11. Cross-section plots for all HIPPO, ORCAS, and ATom campaigns are presented in Figs. S13, S14, and S15.

## 6 Summary

Over the past two decades, we have developed and improved airborne systems for in situ and flask based measurements of atmospheric $O_2$, and have deployed these on a series of 15 regional and global research campaigns. Here we have described the AO2 instrument and Medusa flask sampler to provide support for more detailed scientific studies using their collected data, and with an aim of aiding other investigators who may wish to undertake similar measurements. With this latter goal in mind, we have also detailed the many methodological challenges we have faced in making these high precision measurements and how we have overcome them. Having two independent systems, with the high temporal resolution in situ measurements complemented by flasks sampled at much higher flow rates and analyzed in a controlled laboratory environment, has been critical for detecting and resolving problems in either system. Also, having measurements of $\delta(Ar/N_2)$ on the Medusa flasks has been invaluable for ruling out or detecting and correcting for potential fractionation effects.



The primary sources of potential biases in airborne measurements of, or sampling for, atmospheric $O_2$ concentrations include 1) fractionation of $O_2$ relative to $N_2$ at aircraft inlets (Sect. 4.2) or flask outlets (Sect. 4.1) owing to pressure or temperature driven diffusion, respectively; 2) surface adsorption and desorption effects resulting from drying out of regulators and tubing (Sect. 4.3) or 3) changes in the pressure and humidity ramping of inlet tubing and components on ascent versus descent (Sect. 4.4). These effects may also be important for airborne measurements of $CO_2$ and other gases, but at a smaller absolute level. An additional $O_2$ measurement concern unique to the use of intense VUV radiation in the AO2 detector appears to be the presence of varying concentrations of residual water vapor or hydrocarbons potentially leading to photochemically-induced changes in optical window transmission (Sect. 4.5). As described above, we have taken measures to mitigate these potential biases, and when necessary filter or empirically correct for them, such that they do not adversely influence scientific interpretations of the measurements.

For AO2, we report a $\delta(O_2/N_2)$ precision of $\pm$ 2–4 per meg in 5 s for sample air in flight, depending on aircraft motion, and $\pm$ 1.25 per meg in 5 s for calibration gas on the ground (Sect. 2.2). Comparisons between Medusa and ground stations, and between AO2 and Medusa, show no statistically significant bias for Medusa relative to laboratory scales averaged over all global campaigns and no statistically significant bias for AO2 averaged over the 6 most recent campaigns (Sect. 4.7). For all global Medusa campaigns and the most recent 6 AO2 campaigns, campaign-mean offsets from stations and between AO2 and Medusa are all within 5 per meg. For both AO2 and Medusa, the quality of the measurements have improved steadily over time as we have learned from past experiences and modified the instruments or procedures.

We will continue our efforts to improve AO2 and Medusa along several paths. Most helpful for AO2 would be increasing the sample flow rate by a factor of 2 or more, which we anticipate would reduce inlet fractionation, surface effects, and noise from thermal gradients in the inlet cryotrap. However, this will require further development to maintain the high degree of pressure control and adequate drying at these higher flow rates, and the desire for reduced biases would need to be balanced against the drawbacks of a higher flow rate, such as more rapid filling of cryotraps and faster consumption of calibration gases. It may be possible to split the flow to allow for higher inlet flows and lower detector flows, but this would require research on how to eliminate or maintain constant fractionation at the split. The noise contribution from the inlet cryotrap might also be ameliorated with a smaller trap volume, improved flow and pressure control, or valves producing less of a transient flow pulse. It may also be possible to improve AO2 sample air drying by moving the first stage cryotrap to immediately downstream of the inlet control valve or increasing the pressure at the second stage trap, though these steps will also require development work to maintain fine pressure and flow control. Further research is also warranted on inlet fractionation, using high speed wind tunnel studies, and tubing materials or surface treatments to minimize adsorption effects. For Medusa, an alternative design that packaged sets of flasks with automated distribution valves and motorized stopcocks could greatly reduce the required labor associated with swapping and leak testing flasks in the field, albeit at greater cost.

While airborne measurements of atmospheric $O_2$ come with many challenges, the potential for new scientific insights based on these measurements justifies meeting them. Airborne atmospheric $O_2$ measurements provide unique constraints on carbon cycle and physical climate processes (e.g., Bent, 2014; Nevison et al., 2015; Resplandy et al., 2016; Stephens et al., 2018; Asher et al., 2019; Morgan et al., 2019b). Precise, high resolution, global scale, seasonally resolved, profiling airborne measurements



can observe the impact of biogeochemical land and ocean exchanges at large scales and with high fidelity. Further scientific investigations using AO2 and Medusa measurements are planned, and will be facilitated by the methodological presentation given here.

*Data availability.* Web links and DOIs for collections of individual flight AO2 and Medusa data files for each campaign are provided in the
reference list and Table S2. For AO2, these include 1 Hz AO2 data interpolated from the native 0.4 Hz measurements with both the original measurements and those adjusted to match Medusa (Stephens et al., 2011a, b, 2012, 2013a, b, c, 2017a, 2019). For Medusa, these include measured values on each flask as well as files defining the averaging kernel to use when comparing to 1 Hz data (Keeling et al., 2011a, b, 2012a, b, 2013a, b, c, d, e, f; Stephens et al., 2017b, c; Morgan et al., 2019a). In addition to these individual flight files, several merge products are available, which combine AO2 and Medusa data with state parameters and measurements from other instruments. These include a 1-Hz
merge for START-08 (UCAR/NCAR - Earth Observing Laboratory, 2013), 10-sec and Medusa merges for all HIPPO campaigns (Wofsy et al., 2017a, b), 10-sec and Medusa merges for ORCAS (Stephens, 2017), and 10-sec and Medusa merges for all ATom campaigns (Wofsy et al., 2018). We are in the process of updating all the online individual and merge files in conjunction with this publication.

*Competing interests.*

The authors declare that no competing interests are present.

*Acknowledgements.* We would like to thank the many colleagues who have provided support over the years in the form of advice, brainstorming, data, and camaraderie in the field. In particular we would like to thank the pilots, mechanics, technicians, and other support staff of the NCAR Research Aviation facility and Earth Observing Laboratory, the NASA Airborne Science Program and Earth Science Project Office, and the University of North Dakota and University of Wyoming flight research facilities. We are grateful for test flight support, laboratory preparation of standards and flasks, and conducting flask measurements at Scripps from Sara Afshar, Bill Paplawsky, Adam Cox,
and Shane Clark. We would like to thank the Harvard QCLS and NOAA PANTHER teams for their sharing of START-08, HIPPO, ORCAS, and ATom $N_2O$ measurements, including Steve Wofsy, Eric Kort, Rodrigo Jimenez, Jasna Pittman, Sunyoung Park, Roisin Commane, Bin Xiang, Greg Santoni, John Budney, Yenny Gonzalo Ramos, Fred Moore, Jim Elkins, and Eric Hintsa. We also thank Andrew Manning, Benni Birner, and Yuming Jin for valuable discussions. The measurements presented here have been supported by grants from NASA and DOE to Scripps (COBRA-1999test and COBRA-2000), NASA grants NAG5-11430 and NCC5-590 (COBRA-2003), NSF EAR-0321918
(ACME-07), NSF ATM-0628519 and ATM-0628388 (START-08 and HIPPO), NSF PLR-1501993 and PLR-1502301 (ORCAS), NSF AGS-1547626 and AGS-1547797 (ATom-1), and NSF AGS-1623745 and AGS-1623748 (ATom 2-4). This material is based upon work supported by the National Center for Atmospheric Research, which is a major facility sponsored by the National Science Foundation under Cooperative Agreement No. 1852977.





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





**Figure 1.** Plumbing diagram of the AO2 instrument. AO2 consists of an inlet, a pump box, an insulated cylinder box, an analyzer box, a single cryotrap shown here in 2 parts, and an external purge cylinder. See Sect. 2.1 for a description of the individual components. See Table S1 for selected vendor and part numbers.

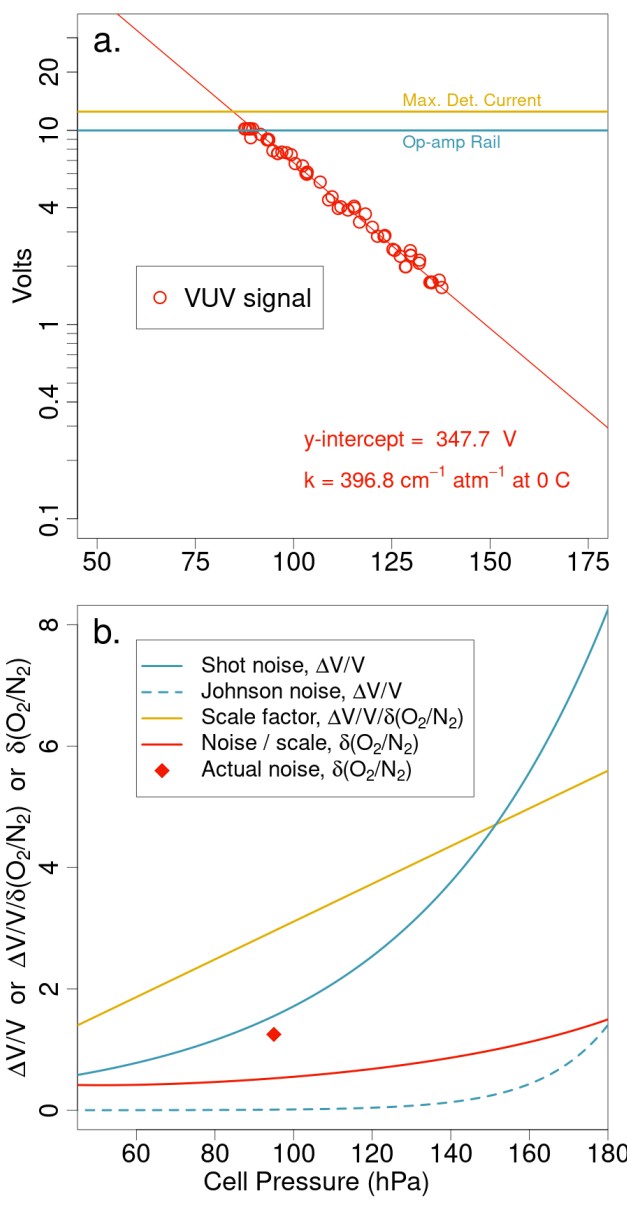

**Figure 2.** Typical results from a pressure scan of the AO2 detector cell showing (a) the logarithmic relationship between detector volts and cell pressure, as well as current limits for the amplifier and photocathode. Owing to sensitive tuning of the cell pressure control system in this configuration, control above 140 hPa is unstable. The values in (a) give the unattenuated lamp signal and apparent absorption coefficient defined by the y-intercept and slope of the fit. Using these values, (b) shows predicted noise contributions to comparisons of subsequent 2-sec averages from shot and Johnson noise, as well as the Beer's law scaling between relative changes in $\delta(O_2/N_2)$ and detector output, and the resulting predicted noise in $\delta(O_2/N_2)$. The single point in (b) corresponds to typical performance while running calibration gas either directly or through a trap at room temperature (Sect. 2.2). There is little change and no minimum in predicted noise over the pressure range shown.

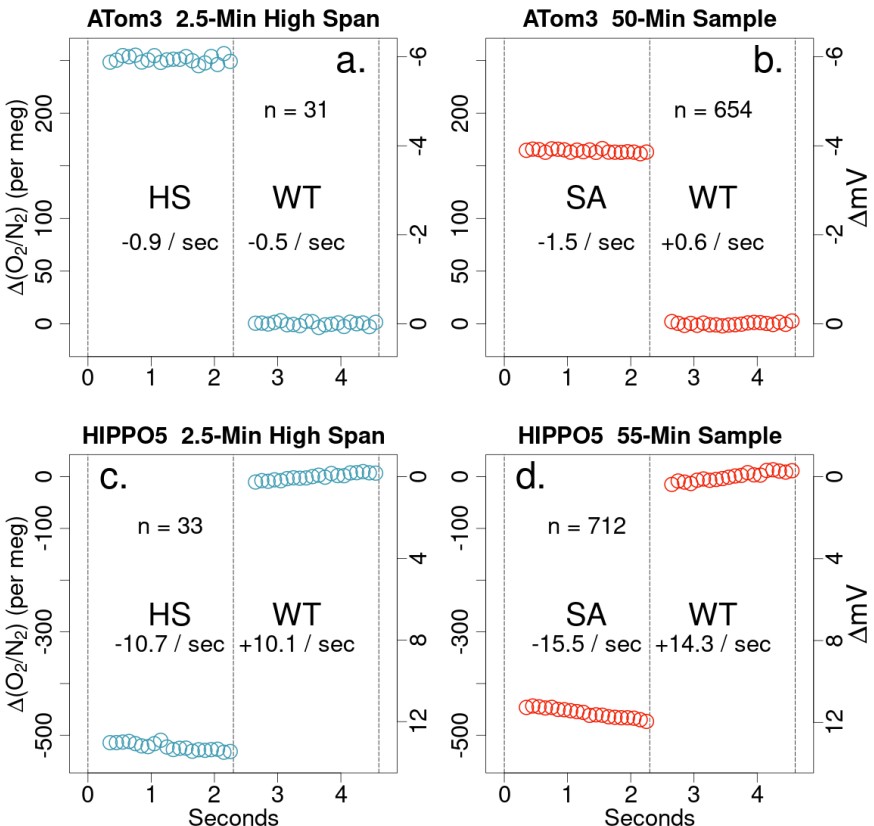

**Figure 3.** Average AO2 square wave shapes from a high span (HS) calibration period (a, c) and sample (SA) gas period (b, d) from an example ATom-3 flight (a, b) and an example HIPPO-5 flight (c, d). Points are calculated as the median VUV signal binned by jog position over multiple jogs, as indicated by the n value in each panel, and plotted relative to the average working tank (WT) signal. Dashed vertical lines indicate the times when the 4-way changeover valve switched. The gaps after switching correspond to the system logging less frequent diagnostic signals and no points have been removed during the transitions. The right y axes show the raw VUV signal in mV and the left y axes show approximate per meg $\delta(O_2/N_2)$. Slopes of the individual span and working tank segments are also reported in units of per meg / sec in each half panel. The difference between span gas and working tank slopes is only -1 to -2 per meg / sec in the ATom-3 example, but -20 to -30 in the HIPPO-5 panel, which we attribute to inadequate drying (see Sect. 4.5).

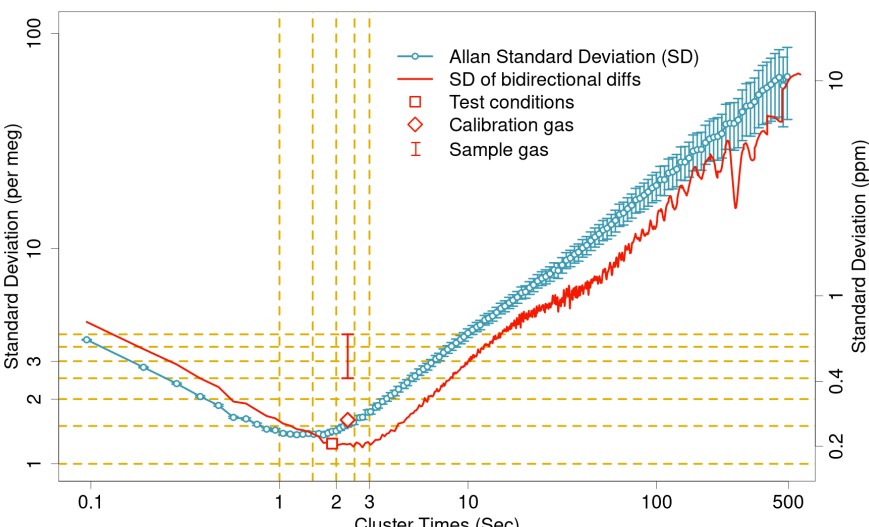

**Figure 4.** AO2 signal noise characteristics from a laboratory test running a single calibration gas without changeover valve switching for 40 min on a log-log plot. The two-sample Allan Standard Deviation and errors are as calculated by the allanvar package in R. The standard deviation of the three-sample bidirectional differences are calculated as in AO2 processing for hypothetical valve switching intervals. During this test the instrument was able to make 20 analog to digital conversions in 1.9 s rather than 2.3 s as in flight, owing to sampling fewer diagnostic signals. Symbols show the results of applying the AO2 processing on 20-sample intervals from this lab test, as well as a more typical value for calibration gas during smooth flight conditions, and a typical range of values for sample gas in flight during smooth to moderate turbulent conditions. The increase in noise for calibration gas in flight is related to slight motion and trends during a calibration cycle. The increase in noise for sample air in smooth flight is likely caused by thermal gradients in the cryo tap. The left y axis shows values in per meg $\delta(O_2/N_2)$ and the right y axis shows ppm in $O_2$ mole fraction.

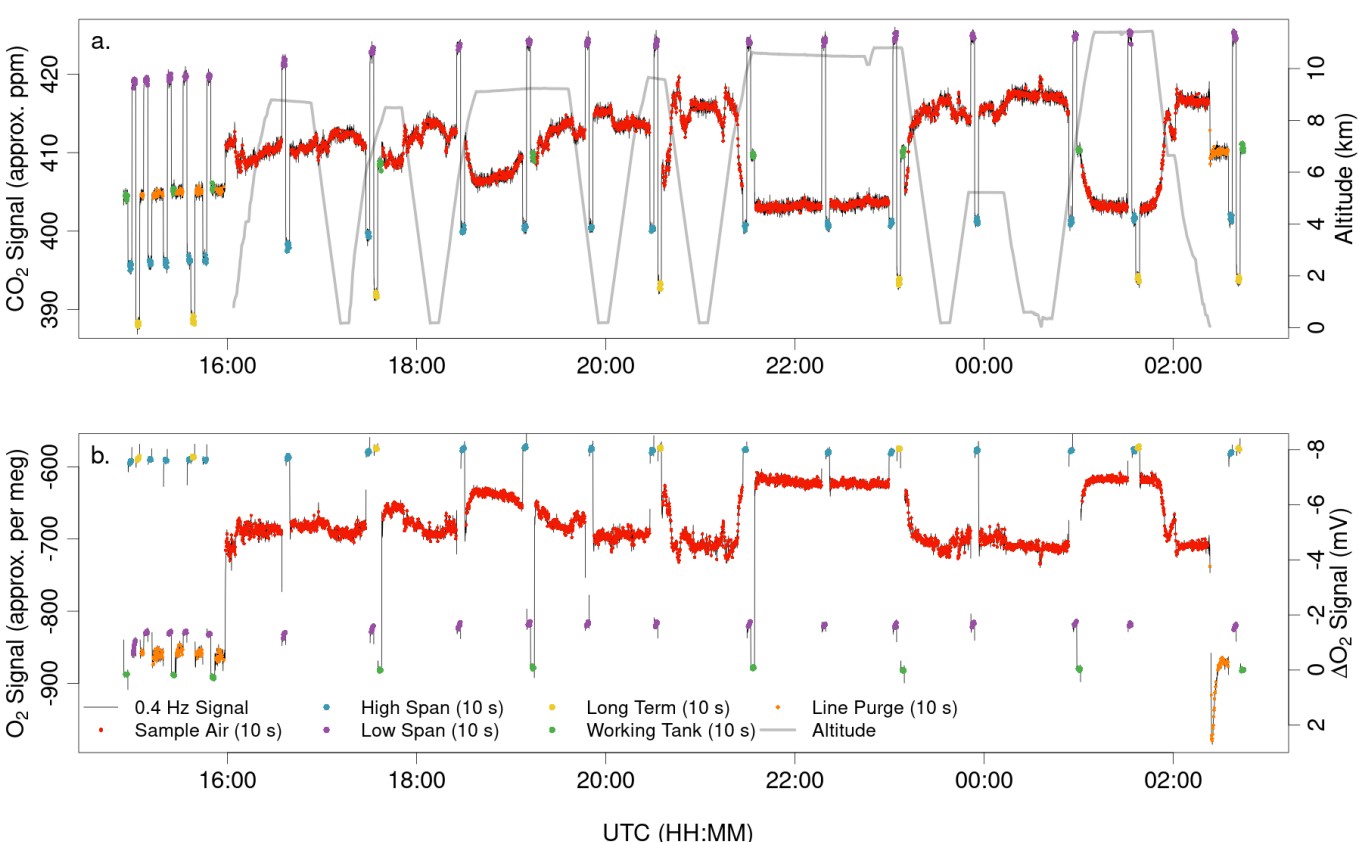

**Figure 5.** AO2 $CO_2$ and $O_2$ signals from ATom4 RF02, from Palmdale, California to Anchorage, Alaska with profiles over the northeast Pacific and Arctic Oceans. $CO_2$ signals are from the interal Li-820 sensor and $O_2$ signals are the amplitude of the 5-s square wave, converted to approximate ppm and per meg using a single linear calibration for the entire flight for each species. The calibration intervals include the last 1 hr 15 min of pre-flight with line purge gas on the sample line and calibration cycles every 15 min; in flight calibrations nominally spaced by 50 min, and 15-min of post-flight line purge and a final calibration cycle. The anticorrelated vertical gradients in $CO_2$ and $O_2$ are consistent with buildup of industrial emissions and respiration over winter and late spring.





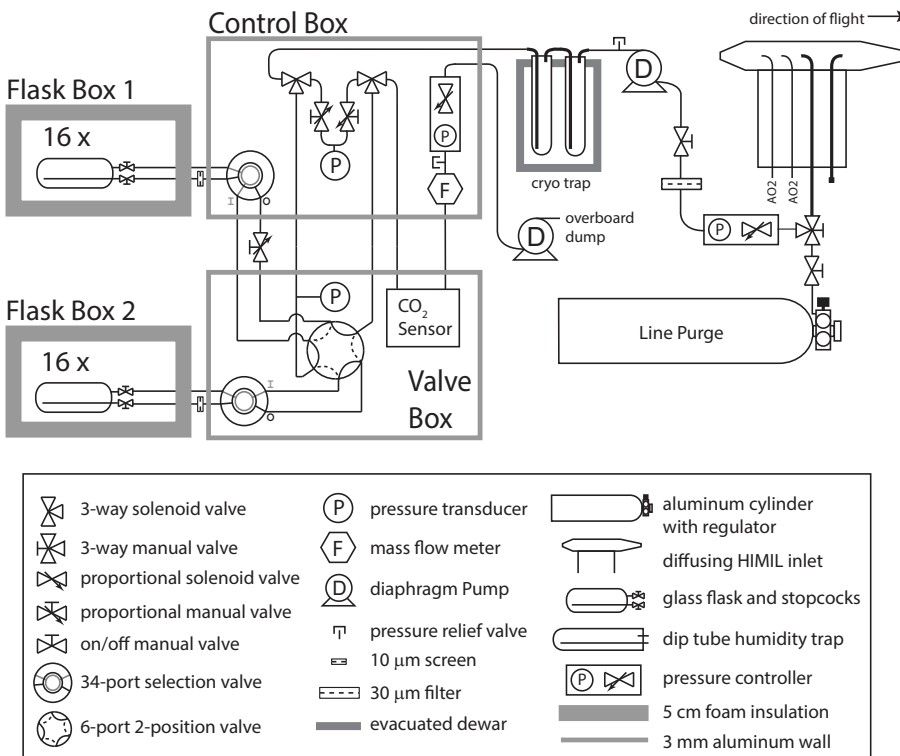

**Figure 6.** Plumbing diagram of the Medusa flask sampler. Medusa consists of an inlet, a control box, a valve box, two insulated flask boxes, a cryotrap, two pumps, and a purge cylinder. See Sect. 3.1 for a description of the individual components. See Table S1 for selected vendor and part numbers.

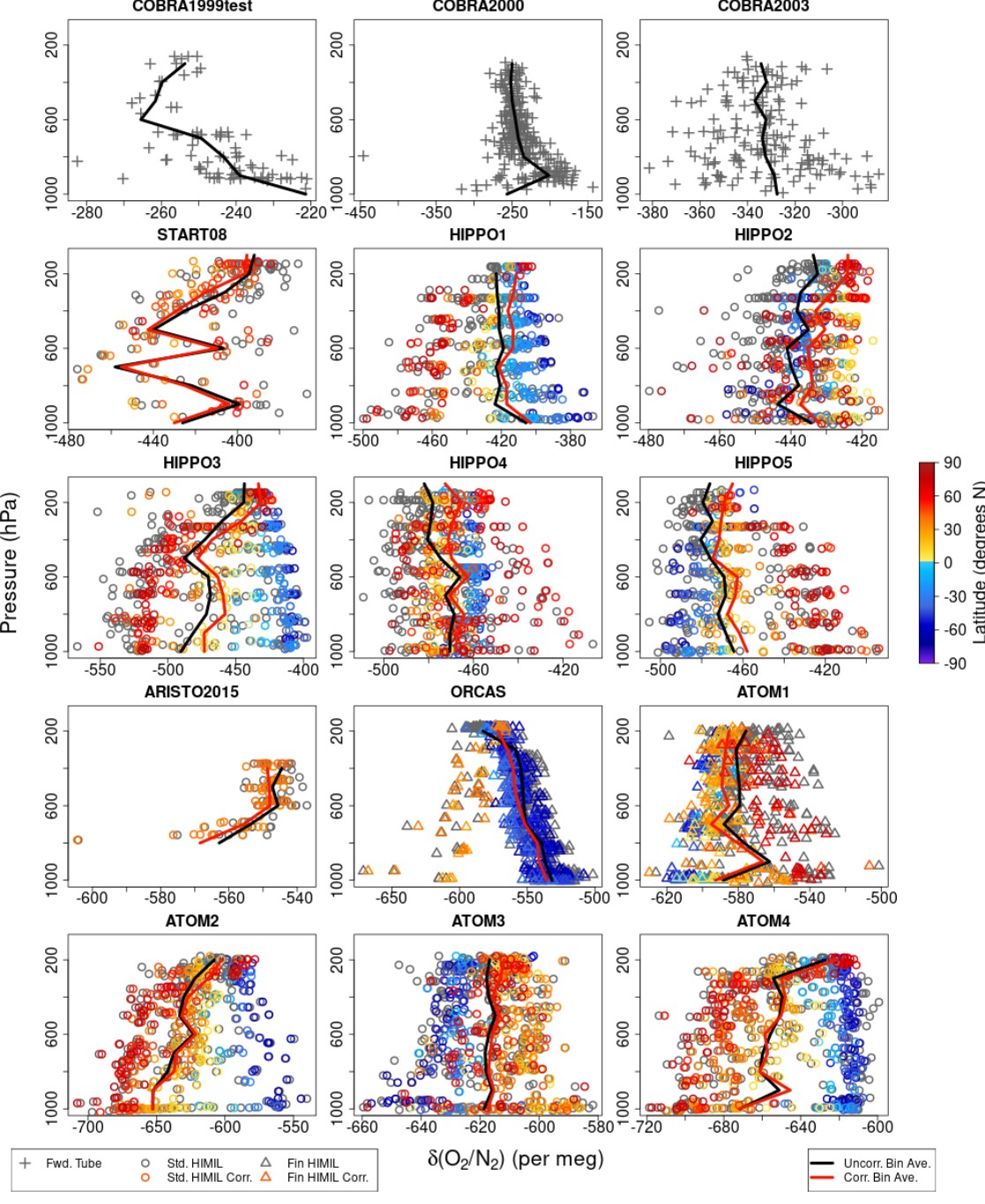

**Figure 7.** Measurements of $\delta(O_2/N_2)$ and $\delta(O_2/N_2)^*$ on Medusa flasks for each campaign plotted versus pressure, and colored by latitude. Symbol shapes distinguish the different Medusa inlet types. Gray symbols show the original uncorrected $\delta(O_2/N_2)$ measurements. Colored symbols show $\delta(O_2/N_2)^*$ data after correction from thermal or inlet fractionation effects using $\delta(Ar/N_2)$. We have not calculated $\delta(O_2/N_2)^*$ for the COBRA campaigns. Black lines show $\delta(O_2/N_2)$ averages for 100 hPa bins. Red lines show bin averages for $\delta(O_2/N_2)^*$.





**Figure 8.** AO2 $\delta(O_2/N_2)$ minus Medusa $\delta(O_2/N_2)^*$ differences for each campaign plotted versus pressure. Symbol shapes distinguish the different AO2 inlet types. Blue symbols show differences between raw unadjusted AO2 $\delta(O_2/N_2)$ measurements and corrected $\delta(O_2/N_2)^*$ Medusa measurements. Yellow symbols show differences after adjusting the AO2 measurements to match Medusa $\delta(O_2/N_2)^*$ by a linear time-of-flight trend plus mean offset for each flight, or in the case of the second half of ATom-1 by a linear pressure trend plus mean offset for each flight. Blue lines show the differences using unadjusted AO2 $\delta(O_2/N_2)$ averaged by 100 hPa bins. Red lines show binned averages using adjusted AO2 $\delta(O_2/N_2)$.



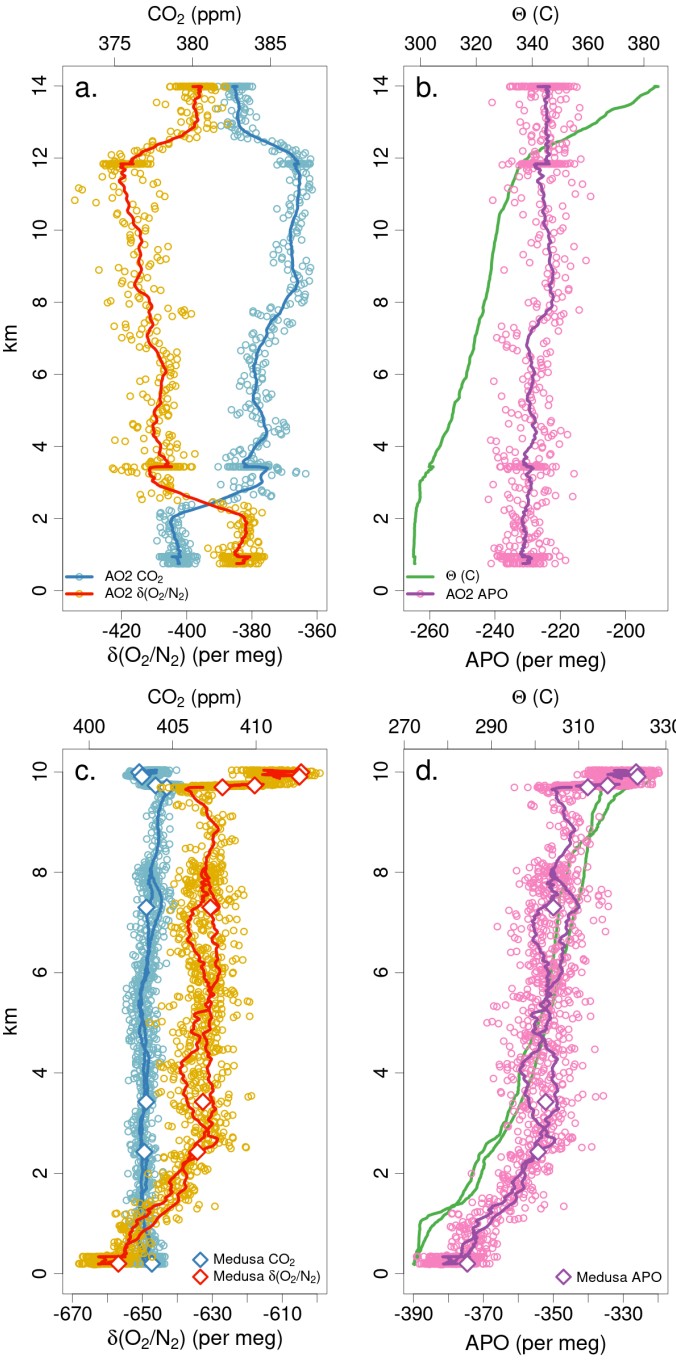

**Figure 9.** Example vertical profiles measured by AO2 and Medusa. a) $\delta(O_2/N_2)$ and $CO_2$ from AO2 on June 23, 2008 during START-08 research flight 15 at 1730 local time, on approach to Grand Forks, North Dakota, a region dominated by agriculture. b) APO and potential temperature from the same profile as (a). c) $\delta(O_2/N_2)$ and $CO_2$ from AO2 and Medusa over the Southern Ocean (63°S, 145°W) on Oct. 11, 2017 during ATom-3 research flight 6. d) APO and potential temperature from the same profile as (c). For the AO2 data, both the 0.4 Hz measurements (points) and 60-sec running means (lines) are shown. In both (a) and (c) the horizontal $O_2$ and $CO_2$ axes are scaled to be equivalent on a molar basis.



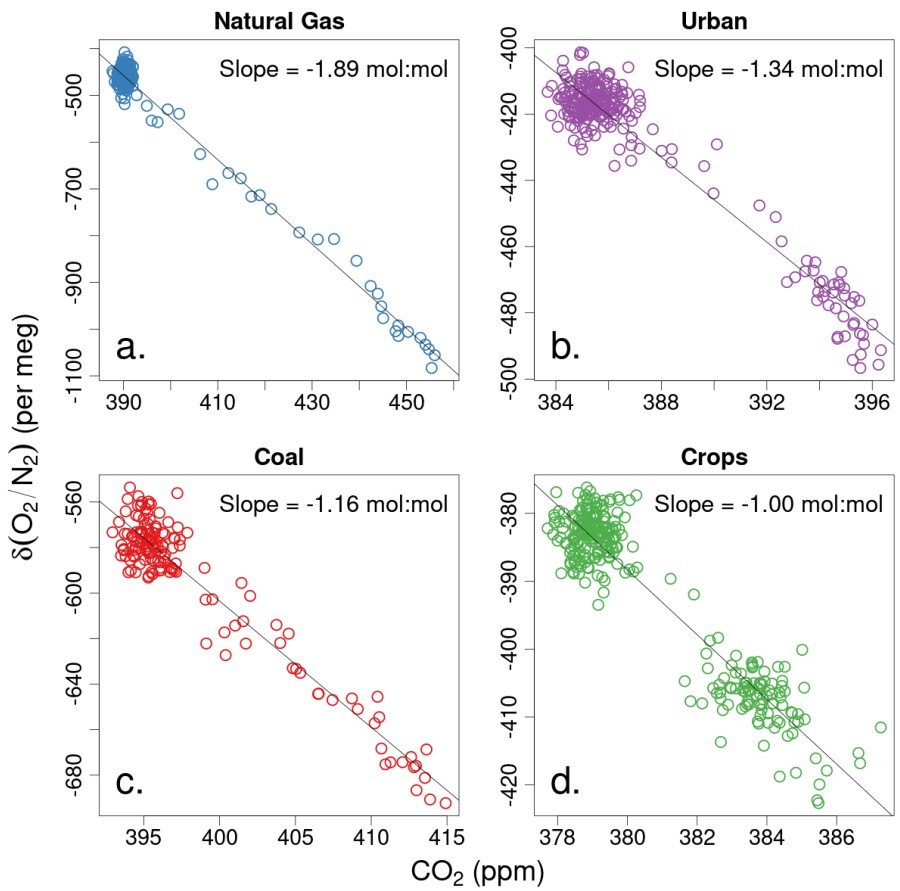

**Figure 10.** Example $O_2$:$CO_2$ relationships observed with the AO2 instrument from sampling a) polluted boundary layer air downwind of a natural gas power plant on approach to Anchorage, Alaska on Jan. 12, 2009 during HIPPO-1 research flight 3, b) polluted boundary layer air on departure from Broomfield, Colorado on Oct. 20, 2009 during HIPPO-2 test flight 1, c) a pollution plume over the San Juan coal power plant near Farmington, New Mexico on June 7, 2011 during HIPPO-4 test flight 1, and d) afternoon boundary layer air on approach to Grand Forks, North Dakota on Jun. 23, 2008 during START-08 research flight 15 (lowest 4 km of profile showin in Fig. 9a). Each panel shows the 0.4 Hz $\delta(O_2/N_2)$ and $CO_2$ measurements and a least-squares fit line with slope reported in molar equivalents.



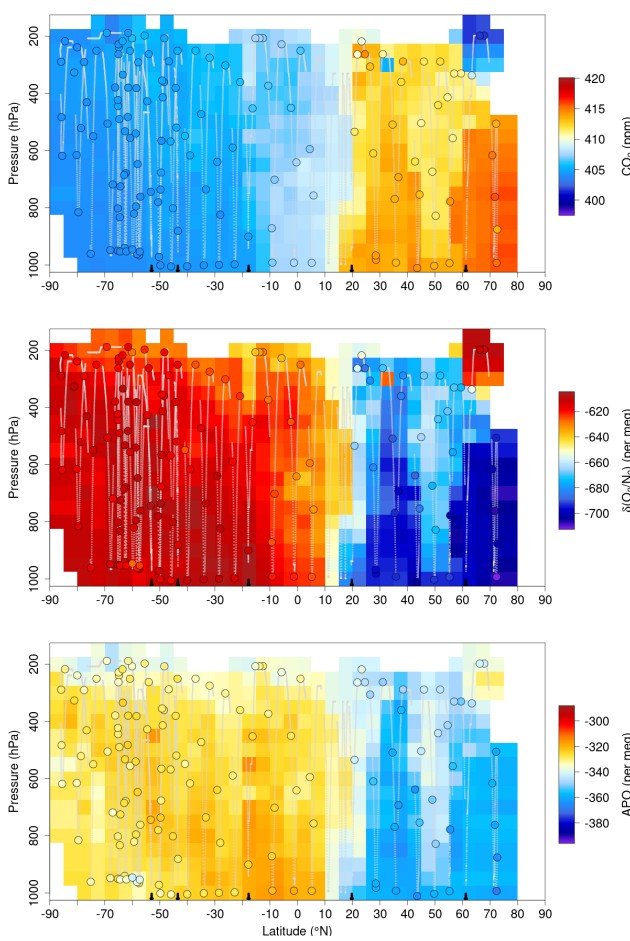

**Figure 11.** Altitude-latitude cross sections from the southbound Pacific transect of ATom-4 for a) $CO_2$, b) $\delta(O_2/N_2)$, and c) APO. Flight tracks are shown as thin gray dotted lines. In situ AO2 data have been interpolated and extrapolated using bicubic spline interpolation with the akima package in R (Akima, 1978) onto a 5 degree latitude by 50 hPa grid. Extrapolation is limited to within 4 degrees latitude and 50 hPa of the observations. Measurements on Medusa flask samples are shown as filled circles. We exclude boundary layer data over land on take-offs, landings, or missed approaches with strong terrestrial influences. Similar plots for all HIPPO, ORCAS, and ATom campaigns are presented in Figs. S13, S14, and S15.