# Peer review of "Airborne measurements of oxygen concentration from the surface to the lower stratosphere and pole to pole"

_Atmospheric Measurement Techniques, 2020_

## Referee Comment (RC1) · Anonymous Referee #1 · 6 Oct 2020

In this paper, the authors describe in detail their techniques to improve airborne system for in-situ (AO2) and flask-based (Medusa) measurements of atmospheric O2. As cited in the Introduction of their paper, some previous studies have reported that the measurements of d(O2/N2) and d(Ar/N2) obtained onboard aircrafts were fractionated significantly from their natural values, and the cause(s) of the fractionation not completely understood. In the present study, the authors have examined some possible causes of the fractionation processes and have succeeded in reducing or correcting for the large fractionation of AO2 d(O2/N2) by using Medusa d(O2/N2)* (d(O2/N2) corrected for thermal fractionation) as reference points. This paper makes a valuable contribution to improving our analyses and interpretation of such datasets as the au-

thors' published HIPPO data, as well as airborne measurements of O2 in general. I have found the paper to be well written and should be published in AMT with a few additional minor revisions identified below.

1) P12, line30: "(Sect. 4.4)," should be changed to "(Sect. 4.4).".

2) P13, lines 20-25: The authors suggest that the thermal fractionation found in Medusa flask is due to the flask dip tube being cold in comparison to the surrounding flask air. However, if continuous flow with a high flow rate of 1,559 to 2,700 ml min-1 is established in the flask, then, I expect, no significant fractionation would occur during the air sampling. I think fractionation occurs during the time period after the rotary valve is isolated and then the flask stopcocks closed (several minutes to an hour). I would like to hear the authors' thoughts on this.

3) P14, lines 1-14: The Medusa d(O2/N2) measurements are corrected for thermal fractionation effects, however, the relationships between the d(Ar/N2) and APO shown in Fig. S5 are not those one would expect from thermal fractionation. Do authors agree? It seems to me that short-term variations in APO due to fossil fuel consumption and/or to air-sea O2 and CO2 fluxes, with the OR being different from 1.1, would constitute bigger contributing factors than the thermal fractionation.

4) P16, line 6: "140 hPa (Fig. S7" should be corrected to "140 hPa (Fig. S7)".

5) P16, line 22: "(Fig. 8" should be corrected to "(Fig. 8)".

6) P18, lines 5-28: I did not know that the preferential adsorption of H2O to stainless steel could prevent O2 adsorption, leading to fractionation of O2 and N2. I think this is a valuable insight, but it seems to be speculative. If the effect is the cause of the temporal decrease in AO2 d(O2/N2) seen in Fig. S9, then the decrease should be attenuated with time (in other words, negative biases for calibration gases decrease as the drying proceeds). Did the authors examine long-term AO2 measurements in the laboratory to confirm whether this H2O and O2 adsorption effects did in fact attenuate

with time? Some quantitative evaluation of the effect would be helpful.

7) Related to the comment 6) above, I think it may be better to consolidate the scales of vertical and horizontal axes of the figures in Fig. S9. Such changes will make the comparison of the slopes and variabilities easier.

8) P21, line 13: "exlude" should be corrected to "exclude".

9) P22, lines 9-11: I think CH4 concentration decreases rapidly with increasing altitude in the stratosphere. Does this altitudinal decrease of CH4 affect the observed AO2 d(O2/N2) in the lower stratosphere, due to the hydrocarbon effect suggested by the authors? Also, considering the authors' discussion, it seems to me that sufficient drying of sample air is vitally important for the VUV absorption detector for O2. How dry does the air have to be? It would be quite helpful if the authors could provide the reader with information like, "lower than xx ppm." Such information will be helpful for the researcher who hopes to employ the authors' VUV technique for high time resolution measurements of the atmospheric O2 concentration.

10) P25, lines 2-4 and summary: The authors repeatedly mention that the larger fractionation found in AO2 than in Medusa will be reduced by increasing the sample flow rate. I do agree with this. But I also assume that the authors are aware of the method used in Yamagishi et al. (2008, https://acp.copernicus.org/articles/8/3325/2008/) to increase flow rate at the air intake and split the air with no significant inlet fractionation. A similar method is being used in some continuous observations of d(O2/N2) by Japanese institutes. I remember that Stephens et al. (2007) also developed a special tee configuration for this purpose. I would be very much interested in hearing from the authors about this method.

11) Figure 8 and Table S3: The authors have made significant effort in correcting for various fractionation processes; I think this has inevitably made the correction methods appear somewhat complicated. Do the blue symbols in Fig. 8 show delta_d(O2/N2) with no correction for the AO2 data, or after the ascent-minus-descent adjustment?

[Figure]

Do the yellow symbols indicate delta_d(O2/N2) with all the corrections for the AO2 data (ascent-minus-descent adjustment, time-of-flight correction to match Medusa, and pressure correction to match Medusa)? Is it correct to assume that the "raw AO2 d(O2/N2) minus Medusa d(O2/N2)*" data and the "corrected AO2 d(O2/N2) minus Medusa d(O2/N2)*"data in Table S3 correspond to the blue and yellow symbols in Fig. 8, respectively? If that is the case, then the blue symbols are not "raw unadjusted" AO2 d(O2/N2) measurements... as stated in the Fig. 8 caption. Also, slight altitudinal decreases of delta_d(O2/N2) by about 5 to 7 per meg can be seen in the binned average vertical profiles (red lines) obtained from the Atom2, Atom3, and Atom4 campaigns. Can the authors explain the cause of these altitudinal decreases? Since it is my impression that the observation methods have improved very much from the early START08 campaign to the recent Atom4 campaign, it should be possible to discuss such slight altitudinal decreases.

---

## Referee Comment (RC2) · Anonymous Referee #2 · 8 Nov 2020

This is a very detailed, and most impressive, account of the experiences the groups made on sample and measurement strategies for atmospheric O2. The level of meticulousness is rarely seen. To quote the first sentence of the conclusions: "Over the past two decades, we have developed and improved airborne systems for in situ and flask based measurements of 25 atmospheric O2" You can say that again! "and with an aim of aiding other investigators who may wish to undertake similar measurements."

Many other groups in the field must humbly recognize that they have not advanced to this level of diligence. And for those groups planning to enter the field, it is a clear demonstration of how difficult and demanding atmospheric O2 measurements at the

10-6 level really are.

I recommend publication in AMT.

As for the contents, I found the paper very clear, and easy to read (although of course the great level of detail requires most careful reading). I only have a few questions/remarks, that the authors might want to address:

Page 11 why replace the 150 ml of air during / after CO2, O2/N2 and Ar/N2 but before CO2 extraction? Is this to avoid possible increased permeation until the next measurement? And: A subsample of 90 ml means a very small amount of CO2 for stable isotope analysis, and surely for 14C. I think you mean this is for the second CO2, O2/N2, Ar/N2 measurement? I expect that for 13C and certainly 14C all remaining CO2 will be extracted? Please change the text to make this clear(er).

PAge 17, lines 8-9 "It was designed to reduce the well-known tendency of aircraft inlets to differentially sample heavy and light aerosol particles (e.g., Belyaev and Levin, 1974), a similar effect to our observed separation of heavy versus light molecules."

I have my doubts if these effects are really similar (particles floating in air vs the air molecules themselves), but as the design apparently also serves your goals, something similar must exist...

The title of 4.2.1 For a while I thought that you were going to discuss some kind of physical filter in the inlet of the system, but it is a data filter. I would add that in the title of the paragraph.
* * *

---

## Author Comment (AC1) · 6 Dec 2020

"Airborne measurements of oxygen concentration from the surface to the lower stratosphere and pole to pole" by Britton B. Stephens et al.

Response to Referee #1

Referee comments in *blue italic*, responses in black.

*In this paper, the authors describe in detail their techniques to improve airborne system for in-situ (AO2) and flask-based (Medusa) measurements of atmospheric O2. As cited in the Introduction of their paper, some previous studies have reported that the measurements of d(O2/N2) and d(Ar/N2) obtained onboard aircrafts were fractionated significantly from their natural values, and the cause(s) of the fractionation not completely understood. In the present study, the authors have examined some possible causes of the fractionation processes and have succeeded in reducing or correcting for the large fractionation of AO2 d(O2/N2) by using Medusa d(O2/N2)\* (d(O2/N2) corrected for thermal fractionation) as reference points. This paper makes a valuable contribution to improving our analyses and interpretation of such datasets as the authors' published HIPPO data, as well as airborne measurements of O2 in general. I have found the paper to be well written and should be published in AMT with a few additional minor revisions identified below.*

We thank the reviewer for their time and their helpful comments.

*1) P12, line30: "(Sect. 4.4)," should be changed to "(Sect. 4.4).".*

Fixed.

*2) P13, lines 20-25: The authors suggest that the thermal fractionation found in Medusa flask is due to the flask dip tube being cold in comparison to the surrounding flask air. However, if continuous flow with a high flow rate of 1,559 to 2,700 ml min-1 is established in the flask, then, I expect, no significant fractionation would occur during the air sampling. I think fractionation occurs during the time period after the rotary valve is isolated and then the flask stopcocks closed (several minutes to an hour). I would like to hear the authors' thoughts on this.*

This is a good point, and has motivated us to look into it more closely. Before HIPPO-4, all flasks were closed at the end of the flight, resulting much longer times open to these tubes than in subsequent flights. We have added notes in Table S2 to clarify this. Further, from HIPPO-4 on, we recorded the time of flask closure by hand in flight. We have now transcribed these times and examined $\delta(Ar/N_2)$ as a function of time until closure. This figure shows results for ORCAS and ATom for flasks below 6 km:

[Figure]

We might expect a dependency of $\delta(Ar/N_2)$ on the amount of time the flasks were open to these tubes with no flow, if thermal diffusion during this time was responsible for the scatter. Since we do not see any dependency, nor a major difference in scatter in Figs. S4, S5, and S6 between HIPPOs 1-3 with up to 7 hours until closure and HIPPOs 4-5 with much shorter closure times, we suspect this is not the cause. We have added a sentence at the end of this paragraph:

"We have examined the dependency of $\delta(Ar/N_2)$ on the length of time between isolating a flask with the rotary valve and manually closing the stopcocks and do not find any relationship, suggesting fractionation is not occurring during this time."

*3) P14, lines 1-14: The Medusa d(O2/N2) measurements are corrected for thermal fractionation effects, however, the relationships between the d(Ar/N2) and APO shown in Fig. S5 are not those one would expect from thermal fractionation. Do authors agree? It seems to me that short-term variations in APO due to fossil fuel consumption and/or to air-sea O2 and CO2 fluxes, with the OR being different from 1.1, would constitute bigger contributing factors than the thermal fractionation.*

The referee is right that plotting $\delta(Ar/N_2)$ vs. APO is not necessarily the best way to identify fractionation, owing to the natural variations in APO. However, except in the lower stratosphere, we expect very little natural variability in $\delta(Ar/N_2)$ and with slopes from correlations versus APO less than 2, so we use this figure to rule out large effects. Rather, Fig. S6 plotting $\delta(Ar/N_2)$ vs. the Medusa minus AO2

normalized difference in APO should be a more sensitive indicator of fractionation. In this figure, START-08 and the HIPPO campaigns show evidence of correlations along the expected thermal or pressure slopes. We have added text (*shown in green italics*) to clarify this:

"Fig. S5 shows Medusa flask $\delta(Ar/N_2)$ versus APO for each campaign along with reference lines for expected slopes for thermal and pressure fractionation of flask samples at 1 atm (Keeling et al., 2004). *Except in the lower stratosphere, we expect only small changes in* $\delta(Ar/N_2)$, *and with slopes versus APO less than 2*, *so we use this figure to rule out large effects.* Fig. S6 similarly shows Medusa flask $\delta(Ar/N_2)$ versus normalized Medusa-AO2 APO differences, *which we expect to be a more sensitive indicator of fractionation*."

*4) P16, line 6: "140 hPa (Fig. S7" should be corrected to "140 hPa (Fig. S7)".*

Fixed.

*5) P16, line 22: "(Fig. 8" should be corrected to "(Fig. 8)".*

Fixed.

*6) P18, lines 5-28: I did not know that the preferential adsorption of H2O to stainless steel could prevent O2 adsorption, leading to fractionation of O2 and N2. I think this is a valuable insight, but it seems to be speculative. If the effect is the cause of the temporal decrease in AO2 d(O2/N2) seen in Fig. S9, then the decrease should be attenuated with time (in other words, negative biases for calibration gases decrease as the drying proceeds). Did the authors examine long-term AO2 measurements in the laboratory to confirm whether this H2O and O2 adsorption effects did in fact attenuate with time? Some quantitative evaluation of the effect would be helpful.*

This is also a good suggestion. We do observe that the noted increase in calibration gas measurements during preflight is most rapid initially and attenuates over a period of several hours. This effect appears to be more pronounced when the regulators have been stagnant for a greater number of days. We have not done laboratory tests to specifically examine this attenuation, but have planned new tests of humidity effects on various tubing materials before the next AO2 deployment and can include regulator tests as well.

*7) Related to the comment 6) above, I think it may be better to consolidate the scales of vertical and horizontal axes of the figures in Fig. S9. Such changes will make the comparison of the slopes and variabilities easier.*

Good idea, we have made this change.

*8) P21, line 13: "exlude" should be corrected to "exclude".*

Fixed.

*9) P22, lines 9-11: I think CH4 concentration decreases rapidly with increasing altitude in the stratosphere. Does this altitudinal decrease of CH4 affect the observed AO2 d(O2/N2) in the lower stratosphere, due to the hydrocarbon effect suggested by the authors?*

During the campaigns presented here, the lowest stratospheric $CH_4$ observed was approximately 20 % reduced compared to the surface. In comparison, we observe modest effects for gas with 100 % removal of $CH_4$. These effects are noticeable as changes in the square wave slope of approximately 20 per meg s$^{-1}$ but we have not found a clear relationship between these slope differences and measurement biases that would allow a quantitative estimate of the effect in the lower stratosphere. Using AO2 to sample deeper into the stratosphere this effect could be more important and should be quantified with future laboratory tests by measuring cylinders with no $CH_4$ also with a different non-VUV measurement technique. We have added text (*in green italics*) pointing this out:

"We now avoid using commercially sourced gases lacking ambient $CH_4$. *In the stratosphere, ambient $CH_4$ depletion might also lead to biases in AO2 measurements. For the flights presented here, $CH_4$ in the lower stratosphere was only depleted by 10-20 %, but this could be a greater concern deeper in the stratosphere and warrants further laboratory investigation.*"

*Also, considering the authors' discussion, it seems to me that sufficient drying of sample air is vitally important for the VUV absorption detector for O2. How dry does the air have to be? It would be quite helpful if the authors could provide the reader with information like, "lower than xx ppm." Such information will be helpful for the researcher who hopes to employ the authors' VUV technique for high time resolution measurements of the atmospheric O2 concentration.*

Drying to a saturation vapor concentration of less than 1.5 ppm appears to be sufficient to avoid anomalous square wave slopes. Furthermore, dilution errors from $H_2O$ scale as -1/0.78 per meg / ppm, so to keep these below 2 per meg, 1.5 per meg also seems desirable. We have added text (*in green italics*) making these points:

"Also, since HIPPO we have ensured that the traps are drying the air sufficiently, and have adjusted our procedures to avoid introducing wet ambient air into the system when swapping traps between flights. *We find that a saturation vapor concentration of less than 1.5 ppm appears sufficient to avoid anomalous square wave slopes, and also note that this concentration would limit potential $H_2O$ dilution effects to less than 2 per meg. Thus, for VUV measurements we recommend drying to 1.5 ppm $H_2O$ or better.*"

*10) P25, lines 2-4 and summary: The authors repeatedly mention that the larger fractionation found in AO2 than in Medusa will be reduced by increasing the sample flow rate. I do agree with this. But I also assume that the authors are aware of the method used in Yamagishi et al. (2008, https://acp.copernicus.org/articles/8/3325/2008/) to increase flow rate at the air intake and split the air with no significant inlet fractionation. A similar method is being used in some continuous observations of d(O2/N2) by Japanese institutes. I remember that Stephens et al. (2007) also developed a special tee*

*configuration for this purpose. I would be very much interested in hearing from the authors about this method.*

Thank you for reminding us of this nice Ochi-ishi paper. We agree that this method can work, but that careful attention to the design and testing is required. For example, in the case of Stephens et al. (JTech, 2007), the diameter and orientation of the dip tube mattered in lab tests and temperature fluctuations at the pickoff location in the field caused fractionation. So, we have not attempted this approach with AO2 yet, but consider it an option in the future. This would be especially desirable if much greater flows were required, for example in order to make eddy-covariance flux measurements.

*11) Figure 8 and Table S3: The authors have made significant effort in correcting for various fractionation processes; I think this has inevitably made the correction methods appear somewhat complicated. Do the blue symbols in Fig. 8 show delta_d(O2/N2) with no correction for the AO2 data, or after the ascent-minus-descent adjustment? Do the yellow symbols indicate delta_d(O2/N2) with all the corrections for the AO2 data (ascent-minus-descent adjustment, time-of-flight correction to match Medusa, and pressure correction to match Medusa)? Is it correct to assume that the "raw AO2 d(O2/N2) minus Medusa d(O2/N2)\*" data and the "corrected AO2 d(O2/N2) minus Medusa d(O2/N2)\*"data in Table S3 correspond to the blue and yellow symbols in Fig. 8, respectively? If that is the case, then the blue symbols are not "raw unadjusted" AO2 d(O2/N2) measurements. . . as stated in the Fig. 8 caption.*

Thank you for pointing out this confusing presentation. Yes, by "raw AO2" we meant the same thing in both Fig. 8 blue points and Table S3, which is after ascent-minus-descent adjustment but before any Medusa based adjustment. We recognize this is a somewhat arbitrary distinction, but because this adjustment is applied based only on empirical vertical ascent/descent information from the aircraft and in the same software step that applies calibrations, inlet lags, etc. (and without any reference to Medusa) we include it in the category of processing internal to the AO2 system. And yes, the yellow points then include this and the other mentioned corrections. We have added text (*in green italics*) in the Fig. 8 caption to clarify this:

"Blue symbols show differences between raw unadjusted AO2 $\delta(O_2/N_2)$ measurements and corrected $\delta(O_2/N_2)$* Medusa measurements. *Here, by "raw undadjusted" we mean after ascent-minus-descent adjustment but before any Medusa based adjustment.* Yellow symbols show differences after adjusting the AO2 measurements to match Medusa $\delta(O_2/N_2)$* by a linear time-of-flight trend plus mean offset for each flight, or in the case of the second half of ATom-1 by a linear pressure trend plus mean offset for each flight. *The blue and yellow symbols here correspond to the values reported in rows 6 and 9, respectively, of Table S3.*"

*Also, slight altitudinal decreases of delta_d(O2/N2) by about 5 to 7 per meg can be seen in the binned average vertical profiles (red lines) obtained from the Atom2, Atom3, and Atom4 campaigns. Can the authors explain the cause of these altitudinal decreases? Since it is my impression that the observation methods have improved very much from the early START08 campaign to the recent Atom4 campaign, it should be possible to discuss such slight altitudinal decreases.*

This is a good observation. We do not have a definitive explanation for these depletions, but speculate they result from still unresolved inlet fractionation. Since we do not see such effects on HIPPO3-4, we suspect the particular sampling location and conditions on the DC-8 may play a role. We have added a sentence (*in green italics*) on this topic:

"Since moving the inlet location and returning to a diffusing HIMIL for ATom 2-4, we have done further speed tests and switching between inlet sizes and orientations inside the HIMIL tube, and do not observe any signs of inlet fractionation in these tests. *However, we still observe negative deviations of 5-10 per meg in high altitude AO2 minus Medusa $\delta(O_2/N_2)$ differences at ambient pressures < 400 hPa (Fig. 8) during these campaigns, that may still result from AO2 inlet fractionation sampling at this location on the DC-8.*"

---

## Author Comment (AC2) · 6 Dec 2020

"Airborne measurements of oxygen concentration from the surface to the lower stratosphere and pole to pole" by Britton B. Stephens et al.

Response to Referee #2

*This is a very detailed, and most impressive, account of the experiences the groups made on sample and measurement strategies for atmospheric O2. The level of meticulousness is rarely seen. To quote the first sentence of the conclusions: "Over the past two decades, we have developed and improved airborne systems for in situ and flask based measurements of atmospheric O2" You can say that again! "and with an aim of aiding other investigators who may wish to undertake similar measurements."*

*Many other groups in the field must humbly recognize that they have not advanced to this level of diligence. And for those groups planning to enter the field, it is a clear demonstration of how difficult and demanding atmospheric O2 measurements at the 10-6 level really are.*

*I recommend publication in AMT.*

*As for the contents, I found the paper very clear, and easy to read (although of course the great level of detail requires most careful reading). I only have a few questions/remarks, that the authors might want to address:*

We thank the reviewer for their time and for these very nice comments.

*Page 11 why replace the 150 ml of air during / after CO2, O2/N2 and Ar/N2 but before CO2 extraction? Is this to avoid possible increased permeation until the next measurement? And: A subsample of 90 ml means a very small amount of CO2 for stable isotope analysis, and surely for 14C. I think you mean this is for the second CO2, O2/N2, Ar/N2 measurement? I expect that for 13C and certainly 14C all remaining CO2 will be extracted? Please change the text to make this clear(er).*

We replace the air during sampling to avoid fractionation at the flask outlet. In laboratory tests we find that at these low flow rates if the flask pressure is allowed to decrease, the $\delta(O_2/N_2)$ and $\delta(Ar/N_2)$ signals drift with decreasing flask pressure, likely a result of cooling of the flask air inducing thermal gradients at the outlet.

Thanks for pointing out this confusing presentation. We have clarified and added text (*in green italics*):

"This second $CO_2$ analysis is done on a 90 ml subsample without replacement. *We extract all remaining $CO_2$ for the $^{13}C$, $^{18}O$, and $^{14}C$ measurements.*"

*PAge 17, lines 8-9 "It was designed to reduce the well-known tendency of aircraft inlets to differentially sample heavy and light aerosol particles (e.g., Belyaev and Levin, 1974), a similar effect to our observed separation of heavy versus light molecules." I have my doubts if these effects are really similar (particles*

*floating in air vs the air molecules themselves), but as the design apparently also serves your goals, something similar must exist...*

We agree that this is speculative, so have modified the text slightly.

Old text: That the HIMIL design works as well as it does for $\delta(O_2/N_2)$ and $\delta(Ar/N_2)$ sampling is likely attributable to its heritage as an aerosol inlet. It was designed to reduce the well-known tendency of aircraft inlets to differentially sample heavy and light aerosol particles (Belyaev and Levin, 1974), a similar effect to our observed separation of heavy versus light molecules.

New text (*in green italics*):

That the HIMIL design works as well as it does for $\delta(O_2/N_2)$ and $\delta(Ar/N_2)$ sampling *may be* attributable to its heritage as an aerosol inlet. It was designed to reduce the well-known tendency of aircraft inlets to differentially sample heavy and light aerosol particles (Belyaev and Levin, 1974), a *potentially analogous* effect to our observed separation of heavy versus light molecules.

*The title of 4.2.1 For a while I thought that you were going to discuss some kind of physical filter in the inlet of the system, but it is a data filter. I would add that in the title of the paragraph.*

Good point. We have changed "filtering" to "data filtering."